# Sensing multi-directional forces at superresolution using taxel value isoline theory

Huanbo Sun [1,2,3] ✉, Adam Spiers[1,4], Hyosang Lee [1,5], Jonathan Fiene[1] & Georg Martius[1,6] ✉

Robots can benefit from touch perception for enhanced interaction. Interaction involves tactile sensing devices, contact objects, and complex directional force motions (normal and shear) in between. We introduce a comprehensive theory unifying them to advance sensor design, explain shear-induced performance drops, and suggest application scenarios. Our theory, based on sensor isolines, achieves superresolution sensing with sparse units, avoiding dense layouts. Through structural analysis of the sensor perception field, force sensitivity, and contact object effects, we also explore the force direction influences: normal, tangential shear, and radial shear forces. The model predicts an inherent accuracy reduction under shear forces compared to pure normal forces. Validation used Barodome, a 3D sensor predicting contact locations and decoupling shear/normal forces. Its performance confirmed the significant impact of shear forces, with observed drops (0.5 mm) closely matching theoretical predictions (0.33 mm). This theory provides valuable guidance for future tactile sensor design and advanced robotic touch systems.

Humans possess a remarkable ability to interact with their environment through touch, with our fingertips being a prime example. The skin on our fingertips contains multiple types of mechanoreceptors that work together to provide us with rich tactile information[1,2]. Some of these receptors detect stresses arising from interactions between contact objects, soft tissues, and bones (Fig. 1a, b), and exhibit differences in their sensing capabilities based on their morphology and depth in the dermis[3,4]. By understanding the intricacies of these mechanisms, we can develop new technologies that mimic the human sense of touch, such as touch-sensitive bio-inspired fingertips for robotics (Fig. 1a).

Tactile sensing technologies have revolutionized robotic applications by enabling robots to perceive their environment through touch[5-8]. These technologies utilize deformable sensing elements made of functional materials that convert deformation into electrical signals through various transduction methods, including resistance[9-15], capacitance[4,16,17], light intensity[18-20], magnetic flux[21,22], etc. Except in vision-based sensors[19,20], the sensing elements, known as taxels, are typically arranged in a grid pattern across a surface to match the physical resolution. However, increasing the number of taxels for better spatial resolution can lead to a more compact layout with smaller taxels and thinner wires, resulting in decreased temporal resolution and reduced robustness.

Rather than a dense placement, anatomic studies of human skin[23] reveal that receptors are discretely distributed at varying depths and densities to realize a wide sensing area with spatially varying sensitivities. User studies using haptic wearables[24] validate that the spatially variable perception sensitivity coincides with the distribution of receptor density on the human hand. Biologists also suggest that the superposition of several taxel/receptor reception fields achieves

[1]Max Planck Institute for Intelligent Systems, Tübingen, Germany. [2]Yale University, New Haven, CT, USA. [3]Haptic Sensing Lab, Peking University, Beijing, China. [4]Imperial College London, London, UK. [5]University of Stuttgart, Stuttgart, Germany. [6]Eberhard Karl University of Tübingen, Tübingen, Germany. **Summary** Isoline-based theory for tactile sensing explains shear force detection at superresolution. ✉e-mail: huanbo.sun@pku.edu.cn; georg.martius@uni-tuebingen.de

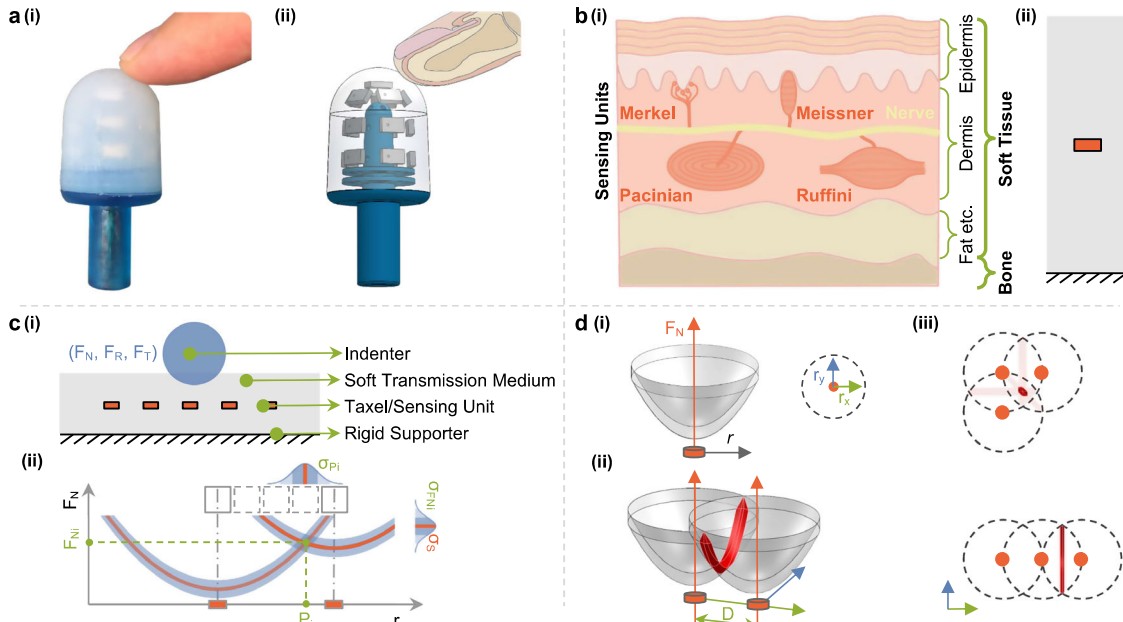

**Fig. 1 | Overview of the theory. a** Real image [a(i)] and schematic drawing [a(ii)] illustrate an adult fingertip interacting with our designed haptic sensor, Barodome, capable of detecting contact-induced pressure changes. **b** Anatomy of human fingertip skin [b(i)] and a simplified model [b(ii)] employed in this study. The model incorporates an orange pressure-sensitive taxel positioned within a gray soft transmission medium fixed on a boundary. **c** Sensor array comprising multiple taxels for sensing contacts with directional forces [c(i)]. Two sensing units, along with their isolines, are utilized to infer contact location ($P_i$) and contact force ($F_{Ni}$), considering uncertainties ($\sigma_{P_i}$, $\sigma_{F_{Ni}}$) introduced by sensor measurement noise ($\sigma_S$) [c(ii)]. **d** Introduction of a single taxel value isoline in three dimensions (3D) [d(i)], two intersected 3D isolines [d(ii)], and three intersected 3D isolines with two different layouts [d(iii)]. The upper layout highlights a practical configuration for localizing contacts with the smallest uncertainty (indicated by the focused red overlapping area).

higher accuracy beyond the physical resolution[25,26]. Recently, Sun et al.[27] proposed a theory to explain the tactile superresolution phenomenon that achieves with a sparse sensor layout a 1000 times higher localization accuracy than the physical density. This innovative approach has made the superresolution-based tactile sensor design strategy increasingly appealing.

Many tactile sensors can achieve superresolution sensing by using receptors fixed on a rigid structure that measures surface deformation. For example, a flexible printed circuit board (PCB) wrapped around a rigid core can measure capacitance changes between sparsely placed pads on the PCB and conductive rubber or fabric insulated by dielectric materials[17]. Alternatively, light-sensitive photodiodes fixed on a hard core can detect reflected light from a coated reflective membrane[28]. In another approach, a camera mounted on the base monitors contact-induced movement of several markers on the internal side of an enclosed hollow shell[18]. Leveraging electromagnetism, a hall-effect sensor fixed on a frame can capture the movement of a permanent magnet floating inside a soft elastomer[29] or detect changes in a magnetic field caused by the movement of a specific encoded magnetic layer sandwiched by a magnetic-free elastomer[22].

These sensors with fixed structures and complex transduction methods are quite different with limited flexibility from the mechanoreceptors in human skin, which are flexibly floating inside soft tissues and simply measure pressure changes. Adhering to the simplicity of our natural system, we propose analyzing the simplest possible setting: a pressure-sensitive sensing unit floating inside soft materials supported by a bone-like structure that can detect both normal and shear forces (Fig. 1). By doing so, we can more accurately mimic the human senses of touch and potentially develop more effective touch-sensing technologies. We draw on the theory proposed by Sun et al.[27] to develop an approach to tactile sensing that includes shear force detection, where most state-of-the-art sensors fall short. Table 1

provides a comparative summary of state-of-the-art tactile sensors with superresolution capabilities. Only a few sensors have been reported to accurately predict contact locations under shear forces. Real contacts are often complex due to unknown object geometries (shape and size) and constantly changing motions (directional contact forces). Thus, in this paper, we aim to understand these complex behaviors and establish a thorough superresolution theory that will facilitate tactile-sensing research from a fundamental perspective.

Our proposed theory, based on taxel value isolines (TVIs), offers an informed approach to building common sensors. It provides guidance on taxel placement within soft elastomers and taxel layout for superresolution-enhanced accuracy; it also provides discoveries about complex behavior introduced by shear contact forces. On the practical side, we follow the instructions derived from our theory to develop a tactile fingertip sensor called Barodome, where the sensing surface forms a three-dimensional cylindrical body and parabolic tip. The overall approach can improve sensor design and manufacturing, data collection and calibration, and is forming a common standard working pipeline.

## RESULTS

### The model

The model is built upon the theory introduced in ref. 27, which considers a class of tactile sensing devices intended for measuring complex directional force interactions on an extended surface (flat/curved) that has an elastic transmission medium embedding physical sensing units (taxels). A 1D model is shown in Fig. 1c. For a point contact, a single taxel value $s$ is a function of the radial displacement $r$ between the contact center and the taxel center, and the applied contact force strength $\vec{F}$:

$$s = f(r, \vec{F}) + \epsilon_S, \tag{1}$$

**Table 1 | Comparison of state-of-the-art super-resolution tactile sensors and Barodome**

| Sensor name | Transduction method | Surface shape | Sensing area [mm²] | # of real taxels | Spacing D [mm] | Spacing of indentation [mm] | Data processing | Normal/shear force | Localization RMSE [mm] | Smpl. SR factor |
|---|---|---|---|---|---|---|---|---|---|---|
| Lepora et al.[17] | Capacitive | 2.5D | ~433 | 12 | 4 | 0.01 | Bayesian Perception | Yes/No | 0.12* (MAE)/– | 798 |
| TacTip[18] | Cam + Marker | 3D | ~2513 | 532 | 4 | 0.01 | Bayesian Perception | Yes/No | 0.1* (MAE)/– | 150 |
| Piacenza et al.[31] | Barometer[32] | 2.5D | 1300 | 5 | ~15 | 1 | Ridge regression | Yes/No | 1.6 (MAE)/– | 32 |
| HapDefX[33] | Resistive | 3D | 24,000 | 10 | ~54 | 2 | MLP | Yes/No | 3.0/– | 85 |
| Piacenza et al.[28] | Optical | 3D | ~6107 | 30 | ~24 | 4 | MLP | Yes/No | 0.6* (MAE)/– | 180 |
| Hellebrekers et al.[21] | Magnetic | 2D | 1600 | 5 (3-axis) | 15 | ~0.8 | MLP | Yes/No | 0.86 (MAE)/– | 46 |
| Yan et al.[22] | Magnetic | 2D | 324 | 9 (3-axis) | 6 | 0.2 | Analytic + MLP + Table | Yes/Yes | 0.1* (MAE)/– | 382[1N] |
| Barodome | Barometer | 3D | 1756 | 16 | 6.5 | 1 | MLP ($P_x,P_y,P_z,F_x,F_y,F_z$) | Yes/Yes for $\forall P_i$ | 0.17[1N]/0.52[1S](0.8[2,N]/1.3[2,S]) | 1209[1N]/129[2N] |

The sign ~ indicates our calculation that approximates missing numbers in the literature.

*RMSE* root-mean-square error, *MAE* mean absolute error, *N* under normal forces, *S* under shear forces

*Indicates the best reported performance. $\Omega = Area/(n \cdot \pi \cdot RMSE^2)$ is used to calculate the super-resolution factor for 2D/3D surface. Smpl. denotes the simple form of $\Omega$ using the reported RMSE.

[1]Evaluation data was not part of the training dataset, but includes positions seen in the training dataset.

[2]Evaluation data does not include any positions that were seen in the training dataset.

where $\epsilon_S$ is the taxel measurement noise with a constant standard deviation $\sigma_S$:

$$\epsilon_S \sim \mathcal{N}(0, \sigma_S^2). \tag{2}$$

One of our major contributions here is to extend the model by considering the significant role of coupled normal ($\overrightarrow{F_N}$) and shear (radial: $\overrightarrow{F_R}$; tangential: $\overrightarrow{F_T}$) forces in complex interactions:

$$\overrightarrow{F} = \overrightarrow{F}_N + \overrightarrow{F}_R + \overrightarrow{F}_T. \tag{3}$$

**Definition 1.** TVIs are a family of curves

$$I^S(r, F_R, F_T) = \begin{cases} F_N & with\ f(r, \overrightarrow{F}) = S \\ undefined & if\ no\ such\ F_N \end{cases} \tag{4}$$

where the mean taxel output Eq. (1) has a constant value $S$, and shear forces always appear along with normal forces in practical applications.

The TVIs for the 1D and 2D model systems and the effect of measurement noise are shown in Fig. 1c, d. The isolines quantify how much normal force is needed to yield the same sensor value. To activate a taxel with a particular value $S$, the required force strength is smaller when the contact location is closer to the taxel, where $F_{N0} = g(s = S)$ represents the pure normal force needed at the taxel center ($r = 0, F_R = 0, F_T = 0$). For a specific sensor reading ($s = S$), there is a one-to-one correspondence between the strength of the normal force ($F_N$) and the contact location ($r$), illustrated by the isolines. The existence of shear forces ($F_R, F_T$) requires additional normal forces to compensate for their impact on taxel values, which in turn disturbs the one-to-one relationship. In this paper, we propose a theoretical model to describe the complex behavior, as summarized by the following equation:

$$I^S(r, F_R, F_T) = F_{N0}(s) + \lambda_{F_s}(s) \cdot \sqrt{F_R^2 + F_T^2} + \lambda_D(s) \cdot (r + \beta_R(s) \cdot F_R)^{\alpha(s)} \tag{5}$$

The isolines are modeled by three terms: The first term $F_{N0}(s)$, captures the force required at zero spatial offset. The second term addresses the effect of the overall shear force modulated by the coefficient function $\lambda_{F_s}(s)$. It describes how much normal forces are needed to compensate for the impact of the shear force on the sensor readings. The third term contains both the effect of the radial distance $r$ and the radial shear force pulling and pushing the elastomer towards and away from the sensing unit, captured with coefficient $\beta_R(s)$. The reduced sensing intensity of distant contacts is modeled by the exponent $\alpha(s)$. In the following, we examine each of these terms individually.

### How is the model derived?

The response from each sensor results from complex interactions between the sensors, objects, and their relative motions. In order to establish a comprehensive framework for future haptic sensor designs, we adopt a simplified model, as depicted in Fig. 1b(ii): a taxel suspended within an elastic transmission medium that gauges pressure changes induced by contact. With this model, our primary focus lies on the sensor's structural characteristics, the size of the contact object, and the directional force interactions (normal and shear) between the sensor and the object. Through these analyses, we anticipate gaining valuable insights into contact behaviors and facilitating the development of innovative haptic sensor designs.

**Structure factors.** In most favorable designs[17,22,28,29], sensing units are mounted on a rigid frame, with soft transmission mediums applied on top. In our study, we adopt a similar approach by enclosing a

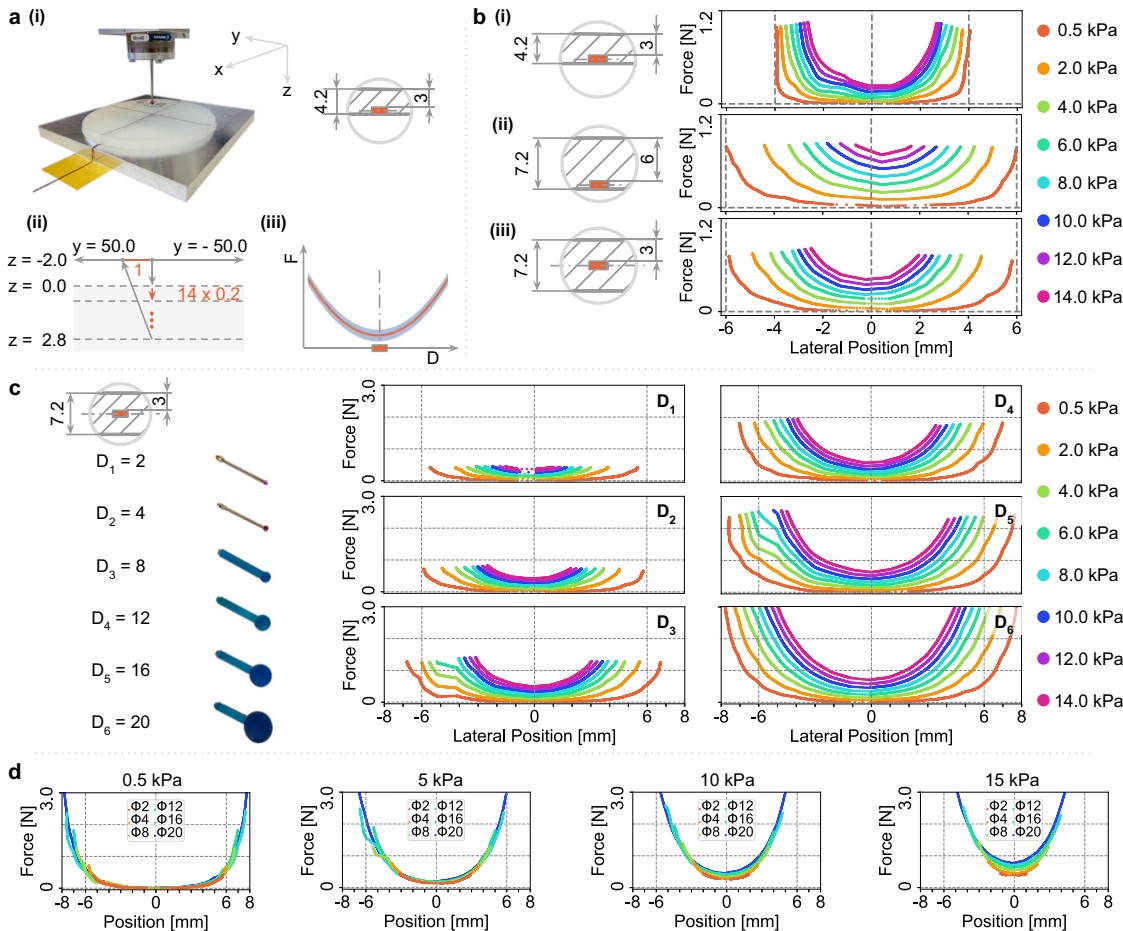

**Fig. 2 | Impact of structure factors and indenter size on sensing unit behavior. a** Experimental setup: a 120 mm diameter elastomer with a 4.2 mm thickness encloses a barometer on the bottom side [a(i)]. The testbed incorporates a four-millimeter spherical indenter, which probes the upper surface of the elastomer along a 100 mm stroke, represented by the black line. [a(ii)] illustrates the data collection trajectory. The Taxel Value Isoline (TVI) is used to quantify the force required to activate the barometer to a specific measurement value along a radial distance [a(iii)]. The blue shaded area represents the sensor measurement noise. **b** Variations in elastomer thickness [b(i), b(ii)] and the relative positioning of barometers within the elastomer [b(ii), (iii)] result in distinct sensing behaviors. **c** Spherical indenters with diameters ranging from 2 mm to 20 mm are employed to probe the embedded barometer. Larger indenters require higher applied forces and can activate the sensor to a certain value further away from the sensing unit center. **d** The impact of varying indenter diameters (Φ2, Φ4, Φ8, Φ12, Φ16, and Φ20) is analyzed at specific sensor readings of 0.5 kPa, 5 kPa, 10 kPa, and 15 kPa.

barometer (MPL3115A2) underneath a piece of elastomer (EcoFlex 00-30) with dimensions of 120 mm in diameter and 4.2 mm in thickness, as illustrated in Fig. 2a(i). To probe the upper surface of the elastomer, we construct a testbed equipped with a four-millimeter spherical indenter, enabling measurements along a 100 mm stroke as indicated by the black line in Fig. 2a(ii). The TVI, depicted as a parabolic "U" shape, characterizes the normal force required to activate the barometer to a certain measurement value across radial distances.

By varying the thickness of the elastomer, we observe changes in the taxel's perception field and force sensitivity, as demonstrated in Fig. 2b(i) and (ii). Increasing the elastomer thickness expands the taxel's perception field while reducing its force sensitivity. This occurs because thicker elastomers are less constrained by boundary effects, allowing for less constrained movements. However, compressing a taxel within a thicker elastomer requires higher forces to achieve the same pressure value, as more material necessitates more energy to generate equivalent local pressure. Additionally, we find that floating the taxel within the elastomer, as shown in Fig. 2b(iii), enhances the force sensitivity due to a local pressure increase towards the surface direction. Depending on specific applications, we can adjust the thickness and sensor depth to achieve the desired perception field size and force sensitivity.

**Object size.** Another crucial aspect to analyze the sensor behavior is its interaction with different objects. In this study, we investigate how the sensor response varies with different object sizes. Based on the above-mentioned observation, we position the barometer in the middle of the soft elastomer to achieve both a large perception field and adequate sensitivity. To assess the sensor's response to various objects, we probe the sensing surface with six spherical indenters with diameters ranging from 2 mm to 20 mm, as illustrated in Fig. 2c.

Smaller indenters require less force to activate the sensor to the same level (Fig. 2d), because they introduce a more localized pressure increase in the sensor readings. Conversely, larger indenters require higher forces, exceeding the maximum capacity of smaller indenters. In such cases, smaller indenters may act like needles, penetrating the soft elastomer and potentially damaging the sensor. Smaller indenters result in a greater sensitivity to force changes, which coincides with the increased sensitivity of human skin to sharp edges. Another significant finding from our analysis is that sensors exhibit distinct sensing behaviors in relation to object size, as shown in Fig. 2d. This suggests that different objects can be distinguished from one another based upon their respective sensor responses. However, it is important to note that a single sensing unit alone cannot achieve this differentiation. By combining multiple sensors into an array, the response pattern

can effectively discern object size[30]. Based on analysis in ref. 27, smaller $\alpha(s)$ values approaching two, and larger perception fields that activate multiple sensors result in higher accuracy. Notably, larger indenters exhibit both characteristics. This finding presents a valuable insight in the field of sensor literature, as it elucidates the mechanism behind the ability to differentiate object size.

**Force direction.** The interaction between sensors and objects significantly affects sensor behavior, particularly in robotic hands where dexterous object manipulation relies on both normal and shear forces. To differentiate between shear and normal forces, independent sensors dedicated to specific directions are commonly employed[22,29]. However, accurately discerning these intertwined motions remains challenging and fails to fully explain the functionality of human skin. Our study aims to make a significant contribution by developing a generalized approach to disentangle force directions through analysis of a simplified setup mimicking human skin.

We include extra shear force components during the interaction in Fig. 3a, in addition to the normal forces depicted in Fig. 2 (see section "Methods for data collection procedure"). These shear forces are analyzed in two directions: the radial direction ($\pm y$-direction), which involves pulling or pushing toward the sensor, and the tangential direction ($\pm x$-direction), which creates a drag force away from the sensor.

Unlike the smooth TVIs observed with pure normal forces (Fig. 3b(i)), the presence of additional tangential shear forces introduces a twisting effect on the TVIs (Fig. 3b(ii)). Note, the visualized TVIs are resultant forces from both normal and shear forces. This disrupts the previous one-to-one relationship, resulting in increased ambiguity in normal force strengths at a single location or ambiguity in location associated with a single normal force strength. Furthermore, the effect becomes more pronounced when the shear forces are in the radial direction (Fig. 3b(iii)).

To gain deeper insights, we decompose the resultant force into normal and shear directions at three shear levels (1.5 N, 0.0 N, and −1.5 N). The decoupled TVIs exhibit smooth patterns. However, the presence of shear forces requires higher normal forces as compensation, as depicted in Fig. 3b(iv) and modeled as $\lambda_{F_S}$ in Eq. (5) (lowest normal forces for different shear force levels). Tangential shear forces ($\pm x$-direction) consistently pull away from the sensor, demanding higher normal forces (Fig. 3b(v)). Radial shear forces ($\pm y$-direction) introduce an offset in the TVIs, while tangential shear forces do not (Fig. 3b(vi)). Radial shear forces either push towards the sensor, adding additional pressure that requires lower normal forces, or pull away from the sensor, requiring greater normal force strengths. For better visualization, we present a three-dimensional representation of TVIs for a sensor reading at 5.0 kPa in Fig. 3c, depicting the contact position (P), shear force ($F_{Shear}$), and normal force ($F_{Normal}$). Notably, Fig. 3c(ii) exhibits a clearly observable slant angle/offset due to the asymmetric radial pushing and pulling effect.

Based on this intriguing finding, we aim to systematically model the behavior of the sensor concerning shear forces. We leverage the definition of TVIs, which exhibit a parabolic shape ($\lambda_D$ and $\alpha$) that varies with sensor readings (S) and shear forces ($F_{Shear}$). For different sensor readings (as depicted in Fig. 3d(i)), both tangential (triangle marker) and radial (circular marker) shear forces demonstrate similar properties, except for the offset ($P_O$) caused by the asymmetric effect of radial pushing and pulling. The offset ($P_O$) represents the location (modeled as $\beta_R$ in Eq. (5)) where the lowest normal force is required for a specific sensor reading under a given shear force condition. TVIs for larger sensor readings indicate a larger $\lambda_D$ with an upper-saturated trend, a smaller $\alpha$ with a lower-saturated trend, a smaller slant angle of the offset ($P_O$) with a lower-saturated trend, and a linearly increased pure normal force ($F_{N0}$ at $F_{Shear}$=0). Importantly, larger sensor readings require higher normal forces to compensate for higher shear forces. All these observations are captured in the model presented in Fig. 3d(ii).

To validate our model for real-world applications involving a mixture of normal, tangential, and radial shear forces, we conduct a quantitative experiment (Fig. 3e(i)). A barometric sensor at ($x = 0$, $y = 0$) interacts with a 16 mm spherical indenter on predefined points. Shear forces are applied along the $x$ and $y$ directions, allowing us to measure tangential and radial shear forces. The offset caused by the shear force direction is modeled using a trigonometric function (Fig. 3e(ii)). Our model in Eq. (5) is evaluated for accuracy, as shown in Fig. 3f. We compare our equation-based model (Fig. 3d(ii) and Eq. (5)) to a lookup table (Fig. 3d(i)), observing similar performance with increasing error beyond a four-millimeter distance (Fig. 3f(i)). The model achieves the highest accuracy at a 45° angle, gradually diminishing towards the sides.

## Tactile superresolution theory modeling considering uncertainty

By considering the structure of the sensor, the size of the object, and the direction of the interaction force, we can accurately model the behavior of the sensor. This model has several implications. Under pure normal force conditions, the TVIs exhibit a bowl-shaped pattern in three dimensions (Fig. 4a(i)). Intersecting two of these TVIs allows for localization of the contact in one direction, with uncertainty ($\sigma_{r_x}$) caused by sensor noise ($\sigma_S$). However, it fails to provide information about the location in the other direction ($\sigma_{r_y}$) and the applied force ($\sigma_{F_N}$)(Fig. 4a(ii)). When three TVIs are arranged in a line, similar issues arise. However, in a grid layout, the location and force strength can be precisely inferred (Fig. 4a(iii)).

Under both normal force and tangential shear force conditions, the TVIs exhibit a four-dimensional nature. For visualization purposes, we present the denture-shaped TVI with three dimensions representing $x$-directional location, tangential force, and normal force (Fig. 4b(i)). Intersecting two of these TVIs allows for localization in the $x$-direction, with uncertainty ($\sigma_{r_x}$), but fails to provide information about the applied normal force ($\sigma_{F_N}$) and tangential force ($\sigma_{F_T}$) (Fig. 4b(ii)). Similar challenges arise when three TVIs are arranged in a line. However, in a grid layout, $x$-location can be inferred, and the normal force and tangential forces can be decoupled. The y-location is not visualized here for clarity. These behaviors also hold for normal and radial shear force conditions (Fig. 4c). Due to the asymmetric effect of radial pushing and pulling (Fig. 4c(i), c(ii)), the uncertainty in all dimensions is larger, even with the grid layout. Logically, if both $x−y$ position, normal, and the two shear forces should be inferred, at least five taxels need to respond, and the accuracy is given by the interaction area of the 5 TVIs. In general, the more sensors respond, the higher the accuracy, but the spatial arrangement matters. The accuracy is limited by sensor noise and the effects of the shear forces, where radial force gives the upper bound (captured by $\beta_R$ in Eq. (5) and Fig. 3e(ii)).

## Theory-informed tactile sensor design

Drawing inspiration from the remarkable tactile sensing capabilities of humans, we develop a touch-sensitive bio-inspired fingertip with promising applications in robotics (Fig. 1a). Human fingertips house pressure-sensitive mechanoreceptors within soft tissues that envelop the underlying bones (Fig. 1a(ii), b(i)). Our sensor, called Barodome, replicates this biological structure to some extent by floating barometric sensor units into a soft silicone material, reinforced by an inner rigid structure (Figs. 1a(ii) and 5a(i)). These design choices were informed by the theoretical analysis mentioned above. More explicitly, we follow a systematic procedure to design and build the sensor prototype, comprising the following steps:

**Step 1: sensing unit selection.** We begin by selecting and analyzing a sensing unit embedded in a soft elastomer. Options include barometric sensors (for pressure), strain gauges (for elongation), and accelerometers (for inclination), as studied in our prior work[27]. We

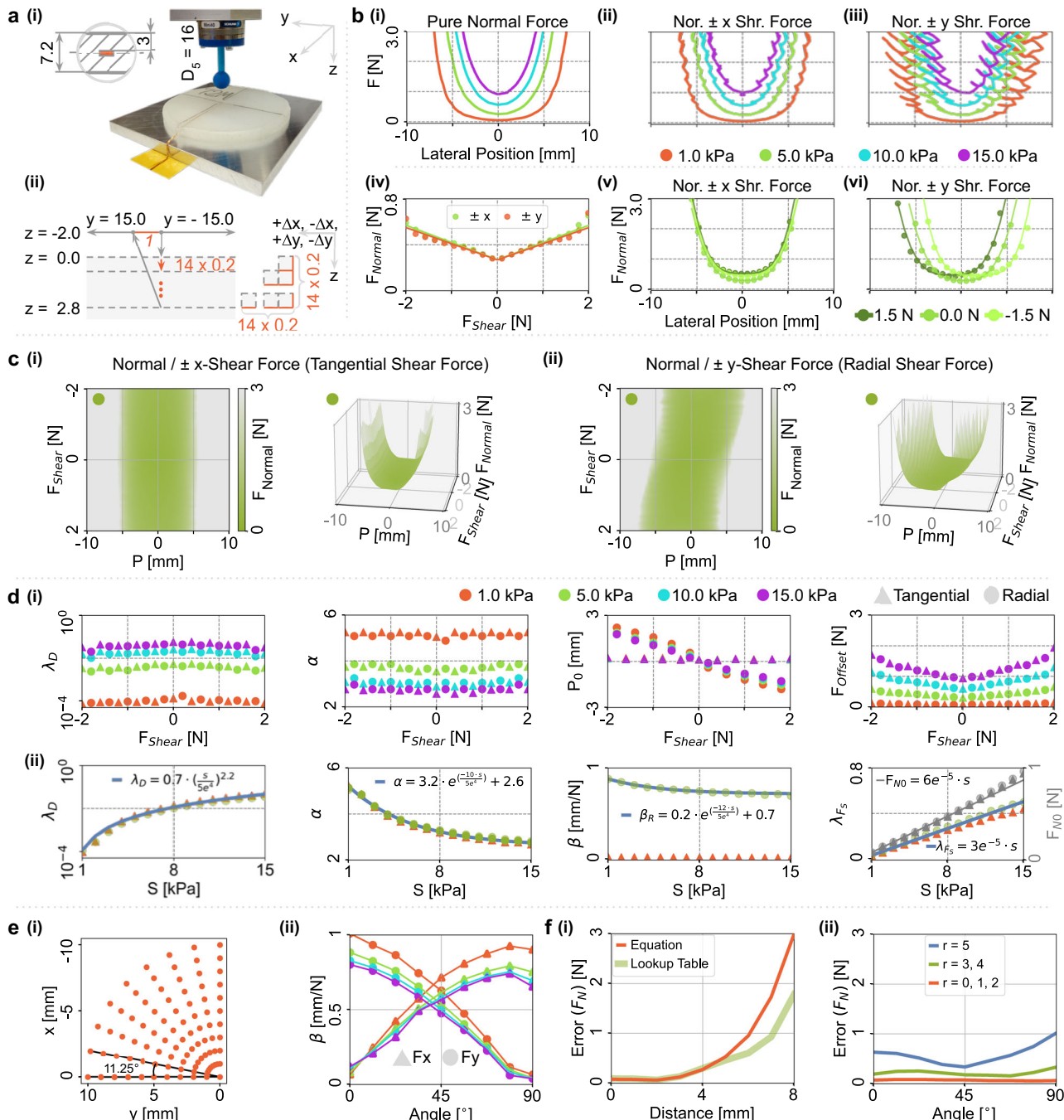

**Fig. 3 | Influence of force directions on TVIs. a** Experimental setup: a 16 mm spherical indenter probes the embedded barometer with both normal and shear forces [a(i)]. The indenter goes along the *y*-axis, spanning a distance from −15.0 mm to 15.0 mm and taking the barometer center as the "zero" position. Depending on the indentation depth, the indenter applies shear forces in four directions (+*x*, −*x*, +*y*, and −*y*) by directional movements separately, along with applied normal forces, as shown on the right part [a(ii)]. **b** shows the isolines for different applied forces: pure normal force [b(i)], normal and ±*x* directional shear force [b(ii)], normal and ±*y* directional shear force [b(iii)], normal and three shear force levels (+1.5 N, 0 N, −1.5 N) along the *x* direction [b(v)], along the *y* direction

[b(vi)], and the lowest normal forces needed for different shear force levels [b(iv)]. All subplots share the same axis except [b(iv)]. **c** shows 3D isoline shape difference between the shear forces along the *x*-axis [**c**(i)] and the *y*-axis [**c**(ii)]. **d** Isoline modeling: [d(i)] shows the parametric modeling of the isolines for different levels of shear forces along tangential and radial directions. d(ii)] shows the parameters' dependence on the sensor value. **e** Experiment of unidirectional shear force indentation: contact points for applying shear forces along *x* and *y* directions [e(i)], and the dependence of parameter *β* on indentation angle between spreading direction and *y*-axis [e(ii)]. **f** Modeling result evaluation along distance [f(i)] and angle [f(ii)].

choose a barometer for its monotonic response and clear force-displacement mapping described by TVIs.

**Step 2: structural design.** We examine the impact of elastomer thickness and the sensing unit's position. Based on performance trade-

offs (shown in Fig. 2), we select a 7.2 mm-thick elastomer with the sensor placed centrally.

**Step 3: material selection.** We assess how material properties (Young's modulus and Poisson's ratio) influence TVIs. Following the

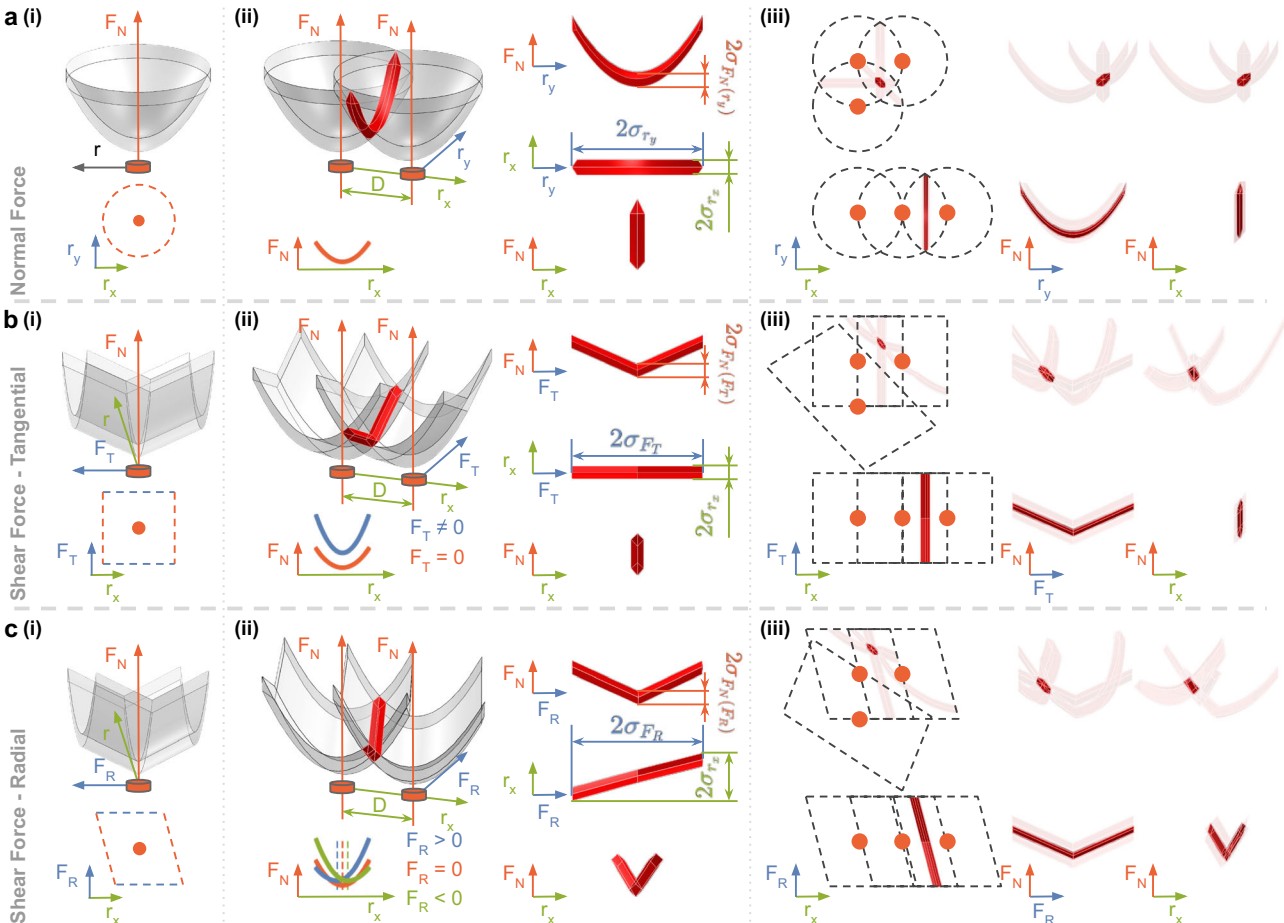

**Fig. 4 | Theoretical modeling of isolines for describing the relationship between normal, shear force directions, and contact positions. a** 3D isoline intersection under pure normal forces. [a(i)], [a(ii)], and [a(iii)] illustrate the shapes and intersection areas of one sensing unit, two sensing units, and three sensing units, respectively. **b** 3D isoline intersection under normal and tangential shear forces. **c** 3D isoline intersection under normal and radial shear forces.

analysis in ref. 27, we use Smooth-On EcoFlex 00-30 (Young's modulus: 0.07 MPa; Poisson's ratio: 0.49999).

**Step 4: indenter geometry evaluation.** We evaluate how different indenter radii affect TVI shape, providing insights into expected performance across interaction scenarios. These indenters represent a wide range of objects that form single-point contact with the sensor. As discussed in ref. 27, superresolution theory enables inference of the resultant force and contact center for single-point contacts. In the case of multiple contacts, they must be separated by at least one taxel spacing to be distinguishable. Continuous (contour) contact scenarios are beyond the current scope of our superresolution model.

**Step 5: shear force effects.** We extend our previous analysis to include shear forces, which are shown to systematically degrade accuracy (Fig. 3). This step reveals the shear-induced shift in TVI shape, essential for predicting real-world performance loss.

**Step 6: noise characterization.** We measure the sensor's intrinsic noise to determine the theoretical accuracy limit achievable via signal processing (Figs. 4, 5d, and 6b).

Steps 1–5 define the TVI shape and its spatial characteristics–perception field, sensitivity, and attenuation profile–all of which govern the achievable resolution and accuracy. These parameters, along with the sensor noise level measured from Step 6, allow us to theoretically estimate performance bounds, as shown in Fig. 2 of ref. 27 and Fig. 6b.

In prior work[27], we demonstrated: (1) a 187-fold theoretical resolution improvement for a 1D sensor, (2) a 106-fold improvement via machine learning, and (3) a 1260-fold improvement for a 2D sensor. In this work, we extend the analysis to a 3D fingertip-shaped sensor (Barodome), focusing on accuracy degradation evaluation due to shear forces. Our model predicts an inherent error of 0.33 mm under shear, and experimental results show a 0.5 mm loss-closely matching the theoretical prediction and validating the extended model.

**Sensor introduction.** The dome-shaped sensor measures 45.0 mm in total height and 21.5 mm in diameter, emulating the size of an adult human thumb tip. The top part serves as the active sensing surface, with a height of 18.9 mm (Fig. 5a(i)). Enclosed within a molded soft elastomer are sixteen suspended barometers, allowing measurement of internal pressure changes upon surface contacts. A 3D-printed rigid support core holds the structure in place. It can accurately report the single-contact location centroid and resultant force magnitude in both normal and shear directions across its spherically symmetrical sensing surface (Fig. 5a(ii)).

**Fabrication.** The barometric sensing unit (MPL3115A2) is directly wired (i.e., without mounting on a PCB) using ultra-thin Cu-enameled wire with a 0.15 mm diameter (Fig. 5b(i)). Sixteen wired barometers are positioned at an equal 6.5 mm distance from their neighbors by a support core, considering the perception field, sensitivity, and barometer geometry (Fig. 5b(ii), (iii)). For assembly, the barometers are molded inside a soft elastomer made of Smooth-On EcoFlex 00-30.

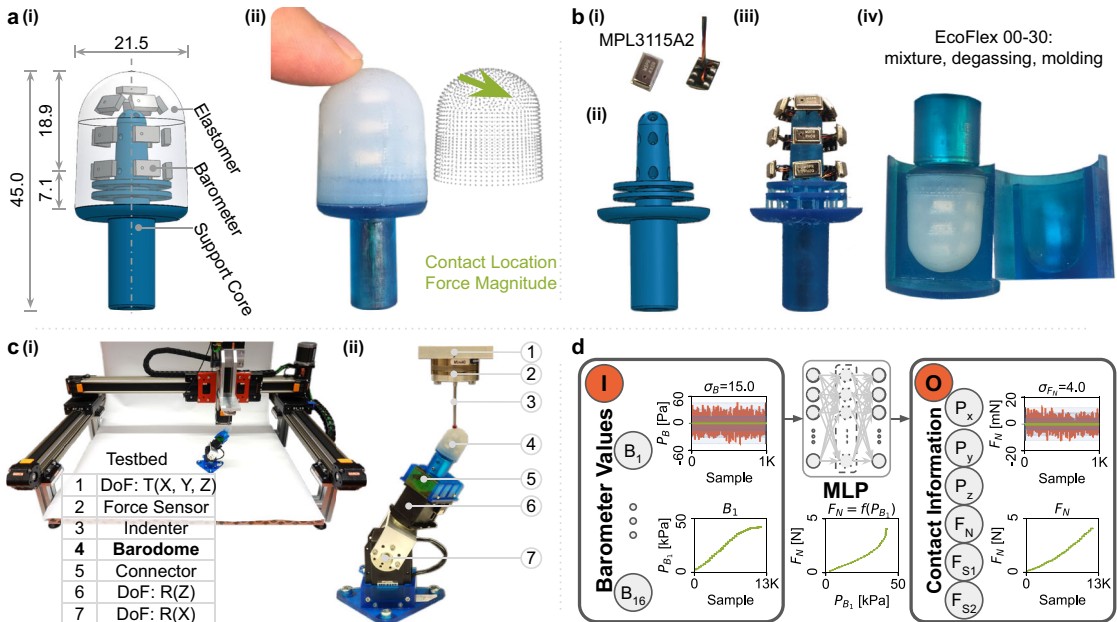

**Fig. 5 | Sensor design flow. a** depicts the sensor design, showcasing the sensor's geometry [a(i)] and the tactile information it provides [a(ii)]. In **b**, the fabrication procedure is demonstrated, including the barometer unit [b(i)], support core [b(ii)], positioning of barometers [b(iii)], and the fully assembled sensor removed from the molds [b(iv)]. **c** showcases the experiment setup, featuring the data collection testbed [c(i)] and a zoomed-in view of the setup [c(ii)]. **d** presents the data processing pipeline: The noise levels of a barometer reading and the force/torque sensor value in the *z* direction (normal force) under no-contact conditions are displayed in the upper-right corner of the blocks. The response of a barometer reading with increasing normal force under contact is illustrated. The machine learning model (MLP) utilizes barometer readings as input to predict position and force values.

The procedure involves material mixing and degassing, followed by molding with a curing time of 4 h. The molds are 3D-printed using Formlabs Form 3 and the material Tough (Fig. 5b(iv)). The fabrication procedure is straightforward and simple.

**Operation pipeline.** The sensor operates on a machine-learning-driven system, utilizing a trained model to predict contact location ($P_x, P_y, P_z$) and directional force magnitudes ($F_N$, $F_{S1}$, and $F_{S2}$) for a single contact based on raw measurements of barometers ($B_1$, ..., $B_{16}$). The barometric sensing unit, with a full measurement range of 90,000 Pa, exhibits a noise level characterized by a variance of 15 Pa, as illustrated in Fig. 5d. To conduct the experiments, we developed a specialized testbed (Fig. 5c(i)) with 3 degrees of freedom (DoF) for translation (Barch Motion Linear Guide has a position resolution of 0.0075 mm) and 2 DoF for rotation (Dynamixel MX-28AT and MX-64 AT result in a maximum position error of 0.03 mm, as shown in Fig. 6a(ii)). This setup carries a four-millimeter spherical indenter to probe the Barodome sensing surface with both normal and shear forces applied. A force sensor (ATI-Mini 40 with a measurement noise level in variance of 4.0 mN) is attached between the probe and the testbed to quantify the contact force magnitudes (Fig. 5c(ii)). A multilayer-perceptron (MLP) structured machine-learning model is trained with a least-square cost function on the collected data that maps raw barometer values to contact information during runtime (Fig. 5d). Model uncertainties as Gaussian residuals stem from barometric sensor noise, testbed accuracy limitations, force sensor noise, and data size constraints. By leveraging the equivalence between minimizing model uncertainty through maximum likelihood estimation and minimizing least-squares error for Gaussian residuals (attributed to barometric sensor noise), the MLP is directly optimized to predict the quantities of interest, as shown in Fig. 1c(ii).

**Evaluation.** We discretize the sensing surface of Barodome into points, spaced 1 mm apart (Fig. 5a(ii)). The testbed uses a 16 mm spherical indenter to probe each point with pure normal forces (Fig. 6a(i)).

Additionally, we collect another dataset with coupled normal and shear forces, similar to the procedure in Fig. 3a(ii). The testbed's spatial position accuracy is <0.03 mm (Fig. 6a(ii)), and it offers a force resolution of 0.01/0.01/0.02 N (Fx/Fy/Fz). Spatial inhomogeneity in coupled forces is due to rigid barometer units and the support core near the base (Fig. 6a(ii)). The mean shear force in the evaluation dataset is 0.43 N (Fig. 6b(i)). Our theoretical model (Fig. 6b(ii)), suggests a 0.43 N shear force leads to an additional 0.4 mm localization error, alongside pure normal forces.

Figure 6 c(i) shows better localization and force quantification accuracy (mean) for pure normal forces than coupled shear forces, matching our theoretical model. With higher indentation forces, both cases improve localization accuracy (more taxels activated) but decrease force accuracy. Limited training data for high forces (histogram in Fig. 6b(i)) or tiny deformation near the support core's bottom base (Fig. 6a(ii)) may explain this trend. High forces lead to sensor saturation and underestimation. The error distributions have zero-centered directional errors (*X*, *Y*, and *Z*), but with long tails from outliers. While localization demonstrates balanced directional accuracy, force quantification exhibits comparatively larger error in radial direction ($F_R$) and notable errors in normal direction ($F_N$). In each resultant force interval, normal forces consistently exceed shear forces in magnitude, while their relative percentage errors remain comparable. Figure 6 c(ii) confirms the superiority of pure normal forces over coupled shear forces, with an accuracy gap of approximately 0.5 mm, consistent with our theoretical prediction of 0.33 mm (dashed blue line). We employ median error and median absolute deviation metrics to mitigate the influence of outliers and non-zero-centered L2-norm resultant errors. As higher forces are applied, localization and force quantification improve, achieving 0.8 mm and 8% for normal forces, and 1.3 mm and 15% for coupled shear forces, with reduced median error and narrower median absolute deviation. When the evaluation data is not part of the training dataset but includes positions seen in the training dataset, the position and force accuracies (root-mean-square error, RMSE) can reach up to 0.17 mm

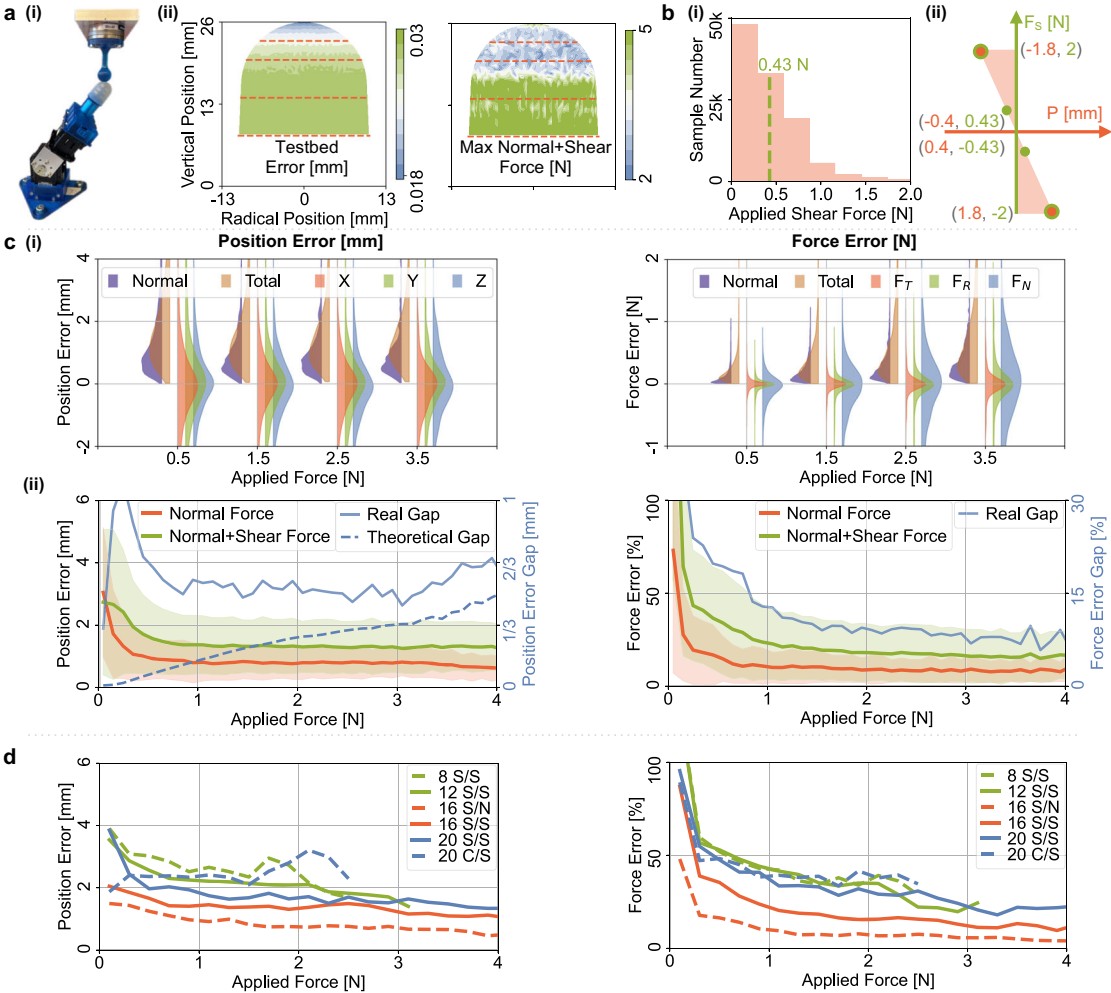

**Fig. 6 | Performance evaluation. a** Experiment setup: The testbed carries a 16 mm spherical indenter to probe the Barodome sensing surface with both normal and shear forces [a(i)]. Testbed accuracy and maximum normal and shear forces are shown in the distribution plot [a(ii)]. The red dashed lines show the locations of barometers. **b** The histogram [b(i)] represents the shear-force component of the total resultant force, while [b(ii)] illustrates the theoretical relationship between localization uncertainty and applied shear force ($F_S$). **c** compares the sensing performance of pure normal forces with coupled shear forces in inferring contact location (left) and force strength (right). Errors are evaluated based on resultant force intervals for both "Normal" (pure normal forces, shown in purple) and "Total"(combined forces including shear components, shown in orange) [c(i)]. For each "Total" force interval, directional position accuracies are analyzed separately for $X$ (red), $Y$ (green), and $Z$ (blue), allowing for comparative assessment. [c(ii)] shows median errors with lighter shaded colors representing the median absolute deviation relative to the applied resultant force. The solid blue line indicates the position accuracy gap between normal force and shear force, and the dashed line is the theoretical mean value of the gap derived from [b(ii)]. **d** Shows the accuracy for different indenter sizes. "8 S/S" refers to an 8 millimeter spherical "S" indenter applying shear "S" forces on the sensing surface. "C" cylinder shape, "N" normal force.

and 1% for normal forces, and 0.52 mm and 4% for coupled shear forces.

Figure 6 d presents accuracy results for various indenter shapes and sizes. The dataset includes four spherical indenters (with diameters of 8 mm−30k samples, 12 mm−30k samples, and 16 mm−35.8k samples of pure normal forces and 179.1k samples of coupled shear forces, and 20 mm−30k samples) and one cylinder shape (with a diameter of 20 mm−10k samples). To simulate realistic scenarios, the dataset contains a large amount of data from one indenter (16 mm) and some data from other sizes and shapes, so that we can assess the performance when applied to less common indenters. The machine learning model is trained on this mixed dataset and exhibits a consistent trend of higher accuracy with increased indentation forces. The 16 mm spherical indenter yields the best accuracy for both pure normal forces and coupled shear forces, particularly in low force intervals. The 12 mm and 20 mm spherical indenters perform comparably, but with slightly reduced accuracy, likely due to their similarity in shape and size and a smaller training dataset.

Conversely, the 8 mm spherical and 20 mm cylinder indenters exhibit decreased performance, likely attributed to larger disparities in their shapes and sizes with others. After approximately 300,000 intensive contact cycles over two weeks, including both normal and shear forces, the Barodome exhibited position accuracy offsets of ($P_x = -0.02$ mm, $P_y = 0.04$ mm, and $P_z = 0.12$ $mm$) and force accuracy offsets of ($F_x = -0.03$ $N$, $F_y = 0.05$ $N$, and $F_z = 0.15$ $N$) for the 20 mm spherical indenter, presumably due to a shift in the sensing units. Further endurance testing is necessary to assess the long-term stability and reliability of the system for various applications, which will be addressed in future work. Future work will focus on mitigating this drift through material improvements and advanced calibration techniques.

## Discussion

Our paper addresses the primary objective of understanding super-resolution sensing in human skin and applying this knowledge to develop advanced tactile sensors (e.g., Barodome) for robots. Through the

proposition of a superresolution theory, we offer simple and easily manufacturable techniques as a practical alternative to complex designs.

The structure of human skin, comprising bones, soft tissue, and mechanoreceptors, plays a vital role in sensing contact. Leveraging this inherent simplicity, we have formulated a theoretical model that incorporates a rigid support core, a soft transmission medium, and pressure-sensitive sensors (Fig. 1b(ii)). This model enables us to comprehensively analyze complex contact behavior, accounting for the interactions between the sensing device and contact object, along with various directional forces (normal and shear) involved in different motions (Fig. 1c).

When designing a sensing device, a primary focus is on high sensing performance using as few physical sensing units as possible. Our theory highlights the significance of a sensing units's perceptive field, sensitivity to force, and sensor noise in achieving this objective, formulating their dependency using taxel value isoline (TVI) (Fig. 2b). By adjusting structural characteristics, such as the thickness of the soft transmission medium and the sensor placement within it, we can customize the sensor's perception field and sensitivity to force accordingly. As concluded in ref. 27, the key properties of the soft transmission medium are Young's modulus and Poisson's ratio. A lower Young's modulus, with similar TVIs, enhances sensitivity but increases deformation under inertia and strong forces. A lower Poisson's ratio (0.5 for elastomers, 0.3 for metals) alters TVIs, reducing radial sensitivity while increasing depth sensitivity.

Analysis of a single sensor unit enables the precise adjustment of hyperparameters in our theoretical model (Eq. 5), ensuring accurate predictions of optimal performance for fully integrated sensing devices, such as a 3D fingertip-shaped sensor Barodome we validated in our experiment. Through a comparison of desired performance goals with the predictions, we can effectively refine the structural and material aspects to achieve our objectives before physically building a whole sensor.

Sensing units exhibit varying reception fields and sensitivities to contacting objects of different sizes and shapes (Fig. 2c). The sensing unit shows higher sensitivity for detecting smaller indenters but with a smaller reception field. Superresolution sensing can effectively reduce costs in applications like touch screens where humans generally have similar finger sizes. However, when detecting objects of different sizes, smaller indenters may require more sensors to account for their narrower perception field. It's essential to avoid applying strong forces that could lead to sensor damage or saturation, considering material tensile strength limits. Complex contact objects can be modeled as combinations of indenters with varying diameters. According to[27], distinguishing multiple contacts requires at least one taxel's distance between them, with at most one taxel activated per contact. Higher contact forces increase the minimum distinguishable distance for double contacts. Spatially continuous contact objects needing detailed force distribution prefer dense taxel layouts, such as arrays or vision-based tactile sensors, as current superresolution-based sensors, limited to resultant force, are inadequate. Additionally, the deformation and contact diameter variations of soft objects during interaction require further study.

One of our paper's key contributions lies in uncovering the significant role of shear forces in sensing performance, posing a challenge in developing tactile sensors to detect and decouple shear forces from normal forces. Multiple combinations of contact locations, normal forces, and shear forces can possibly result in the same sensor reading. Instead of relying on complex sensor structures for direction-specific sensing, we explored analogies to the tactile sensing mechanisms found in human skin. Observations reveal that additional shear forces alter the smooth TVI shape from pure normal forces (Fig. 3b). Distinct TVI shapes correspond to shear force directions (Fig. 3c). Tangential shear forces reduce pressure, decreasing sensor value, requiring higher normal forces for compensation. Torsional shear forces, similar

to tangential shear forces, create a pull-away effect, reduce sensor readings, and demand higher normal forces. It is important to note that torsional shear forces require an offset contact distance to influence the sensor unit effectively. Radial shear forces may pull or push the elastomer, necessitating adjustments in normal forces. This effect is demonstrated in the position dependence with a slant angle (rotational symmetric), as described in Eq. (5) and Fig. 3c(ii).

The slant angle (model in $\beta_R$) systematically decreases sensing performance. In Fig. 4, three sensor units in a grid layout can accurately reconstruct contact location ($P_x$ and $P_y$) and applied normal force ($F_N$) in pure normal force cases, and ($P_x$, $F_N$, $F_T$) in tangential shear force cases. However, the performance of ($P_x$, $F_N$, $F_R$) in radial shear force cases worsens slightly. Despite the potential enhancement from adding more sensors, this deterioration in ($P_x$, $F_N$, $F_R$) remains an irreparable aspect at the system level. Fortunately, our theoretical model enables informed system-level performance predictions, as demonstrated in Fig. 6b(ii).

Based on our findings, we have developed a practical tactile fingertip sensor embedding 16 barometer units within a 3-dimensional cylindrical body and a parabolic tip for robotic applications. Using this sensor, we were able to validate our theoretical model. In normal force cases, the sensor with only 16 barometer units achieves an impressive accuracy of 0.8 mm in localization and 8% in force quantification, covering a force range of 0–4 N. However, we observed a drop in system performance when considering shear forces of 0.5 mm in localization and 7% in force quantification, which matches the 0.33 mm accuracy loss predicted by our theoretical model. This validation reinforces the reliability and accuracy of our theoretical framework in real-world applications.

Our work provides comprehensive guidelines for the design of high-performance tactile sensors capable of accurately sensing shear forces from various aspects. The theoretical modeling not only enhances our understanding of the working mechanisms of human skin but also correlates all aspects of touch sensation, offering valuable insights for engineers to design sensors with improved performance. Additionally, we would like to emphasize that the current theory is specifically developed for single-point contact inference with superresolution capabilities. Scenarios involving continuous or distributed contact patterns, such as those encountered with complex surface contours, fall outside the assumptions of our current model. Addressing such cases would require fundamentally different approaches, such as vision-based tactile sensors[20] or dense array-based tactile systems[4].

As we look to the future, an intriguing avenue for further research involves understanding the influence of different morphologies of the four distinct mechanoreceptors and exploring how their signals can be efficiently merged through a single nerve transmission channel and effectively decoupled on the other end. Additionally, another promising avenue for future exploration is to investigate how to effectively utilize the dense tactile sensing information provided by these sensors. Integrating tactile feedback into the decision-making process of robots holds immense potential for enhancing their autonomy, adaptability, and interaction capabilities in real-world scenarios.

## Methods

Experimental details supporting the findings in this paper are presented below.

### Sensor

To validate the proposed theory, we designed two types of sensors: multiple point sensors and a 3D sensor (Barodome). The point sensor consists of a single barometric unit (MPL3115A2) embedded inside elastomers of different dimensions (diameter: 120 mm; thickness: 4.2 mm and 7.2 mm). The 3D sensor comprises 16 barometric units with 6.5 mm spacing, floating within a dome-shaped elastomer

(dimension: 26 mm in height and 21.5 mm in diameter; thickness: 7.2 mm). To ensure minimal mechanical influence on the taxel, we soldered extra-thin wires (Cu-enameled wire with a diameter of 0.15 mm; ME-MEß Systeme GmbH) to the units, facilitating molding inside the elastomer (Smooth-On EcoFlex 00-30). We designed molds for the sensor spacing and the elastomer molding procedure using SolidWorks 2017. These molds were 3D-printed using a Formlabs Form3 3D printer with Tough material. The units were molded to float in the middle of the elastomer. The molding procedure included a 2-min mixture preparation, degassing of the mixture (5 Pa, 2 min), and molding with a 4-h curing time. We acquired the sensor values of the barometric units (MPL3115A2) through an evaluation board supplied by Adafruit with additional 16-channel analog multiplexers (CD74HC4067). All sensor data was delivered to a laptop (ThinkPad L570) through an Arduino Mega 2560.

## Testbed

We developed a custom testbed with five degrees of freedom (DoFs) to facilitate experimentation. The Cartesian movement of the probe ($\vec{x}$, $\vec{y}$, $\vec{z}$) is controlled by three DoFs using linear guide rails (Barch Motion) with a precision of 0.0075 mm. The orientation of the sensor Barodome (yaw, roll) is set by two DoFs using Dynamixel MX-64AT and MX-28AT servo motors, providing rotational precision of 0.09° and a backlash of 0.33°, resulting in a transitional precision of 0.03 mm at the sensor's tip [Fig. 6a(ii)]. The four millimeter spherical probe is made from an alumimium alloy and is rigidly attached to the Cartesian gantry via an ATI Mini 40 force/torque sensor with a force precision of 0.01/0.01/0.02 N ($F_x$/$F_y$/$F_z$). Other probes are 3D-printed using a Formlabs Form3 3D printer with Tough material. The point sensors and 3D sensors are positioned at the desired orientation, and the probes are used to make contact with them at the desired locations.

## Data collection

We executed the following procedure to achieve a variety of normal and shear forces. The probe moved to a specified location, touched the sensing surface, and deformed it increasingly by moving normal to the surface with a fixed incremental step. For each indentation level, the probe also moved sideways to apply shear forces at a 1:1 normal/shear movement ratio. After a 2-s pause to allow transients to dissipate, we simultaneously recorded the contact location, the indenter contact force from the testbed's force sensor, and the barometric units' values from the molded sensors. All the data were collected and combined using a standard laptop (ThinkPad L570). Specifically, for the point sensors (Fig. 2a(ii)), the testbed made the indenter contact 101 positions evenly spread along the sensor centerline (from $y = -50$ mm to 50 mm) with 14 incremental indentation depths (0.2 mm each) at each position. This procedure applies to all experiments in Fig. 2. For the point sensors (Fig. 3a(ii)), we followed a similar approach, with 31 positions evenly spread along the sensor centerline (from $y = -15$ to 15 mm) with 14 incremental indentation depths (0.2 mm each) at each position. In addition to normal movement, the probe also moved sideways to apply shear forces in either the x or y direction (normal/shear movement ratio 1:1). This procedure applies to all experiments in Fig. 3. For the 3D sensor Barodome (Fig. 6a), we virtually discretized the surface into 1240 points spaced 1 mm apart from each other. For the 16 mm spherical indenter in the pure normal force case, the indenter moved perpendicularly to the surface with 30 incremental indentation depths (0.1 mm each). Likewise, for the 16 mm spherical indenter in the coupled normal and shear case, the indenter moved perpendicularly to the surface with 30 incremental indentation depths (0.1 mm each). For each indentation level, the probe applied shear forces by moving sideways (normal/shear movement ratio 1:1). For experiments in Fig. 6d, the 8 mm, 12 mm, 20 mm spherical indenters, and the 20 mm cylindrical indenter probed only 1/6 of the 1240 points, uniformly distributed.

## TVIs and related parameters

We implemented the following steps to compute the TVIs. Initially, we performed linear interpolation of the sensor values and force values. Subsequently, we selected a sensor value related to position and identified the corresponding force measurement at that specific location. Using the same color scheme as depicted in Fig. 2 and Fig. 3, we then plotted the position-force curve for the chosen sensor value. For isoline derivation, we decoupled normal forces at each shear force level and applied the curve function ($\lambda \cdot r^\alpha + F_{N0}$) to fit each curve. Utilizing these curves, we established a comprehensive lookup table encompassing all possible combinations of sensor values, normal forces, shear forces, and contact locations (Fig. 3d(i)). Furthermore, we employed several curve functions to parameterize these parameters as a function of sensor values (Fig. 3d(ii)).

## Machine learning

We used a standard MLP with ten fully connected hidden layers, each comprising 100 rectified linear units, for all machine learning models. Training involved mean-squared error loss, Adam optimizer (learning rate: $1 \times 10^{-4}$; epsilon: $10^{-5}$), and a batch size of 8192 in 1 million iterations. Separate models were trained for position and force inferences. The data for the 16 mm spherical indenter under pure normal conditions covered 1240 contact locations with 30 incremental steps. They were split into a 3:1:1 ratio for training, validation, and test sets, according to the contact locations. For a 16 mm spherical indenter coupled with shear force, the data covered 1240 contact locations with 30 incremental steps, each having one pure normal force and four shear force directions. The data were split in the same 3:1:1 ratio. For the indenter size and shape comparison experiment, we fixed the test dataset for all the indenters with 40 contact locations, 30 incremental steps, and 5 directions, except for the 20 mm cylindrical indenter with 20 contact locations and 20 incremental steps. The validation and test datasets had the same dimensions. Training datasets for 8 S/S, 12 S/S, 16 S/N, 16 S/S, 20 S/S, and 20 C/S have dimensions of $120 \times 30 \times 5$, $120 \times 30 \times 5$, $1114 \times 30$, $1114 \times 30 \times 5$, $120 \times 30 \times 5$, and $60 \times 20 \times 5$, respectively.

## Data availability

The data and code supporting the findings of this study is available at https://doi.org/10.6084/m9.figshare.28348034.

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

## Acknowledgements

We thank the National Natural Science Foundation of China, the China Scholarship Council, and the International Max Planck Research School for Intelligent Systems (IMPRS-IS) for supporting H.S. G.M. is a member of the Machine Learning Cluster of Excellence, EXC number 2064/1—project number 390727645. We acknowledge the support from the German Federal Ministry of Education and Research (BMBF): Tübingen AI Center, FKZ: 01IS18039B.

## Author contributions

H.S. conceived the theory, methods, and experiments. H.S. derived the theory, designed and constructed the hardware, developed fabrication methods, designed and conducted experiments, and collected and analyzed the data. H.S. drafted the manuscript and revised it. G.M. supervised the derivation of the theory, data collection, and analysis. All authors involved in the initiation of the project, focusing on the design of Barodome. G.M., A.S., H.L., and J.F. revised the manuscript. AI-powered language tools were used on a sentence level.

## Funding

## Competing interests

H.S., G.M., H.L., A.S., and J.F. are listed as inventors on one PCT patent (US20230306261A1). A.S., H.S., G.M., J.F., and E.S. are listed as inventors on one PCT patent (WO2022111798A1) filed internationally that covers the fundamental principles and designs of Barodome.
