## [Transparent Peer Review file · Nature Communications]

Sensing multi-directional forces at superresolution using taxel value isoline theory

Corresponding Author: Professor Huanbo Sun

Version 0:

Reviewer comments:

Reviewer #1

(Remarks to the Author)

The paper proposes a theory based on sensor isolines to guide the analysis of how sensor structure, contact object size, and force components (normal and shear forces) influence sensor characteristics, such as the perception field, force sensitivity, and system errors. The theory is primarily developed for tactile sensors in which multiple sensing units (or taxels) are embedded within an elastomeric membrane, mimicking a human skin-like structure. In this work, barometers are used as the sensing units for a proof-of-concept. On the practical side, the paper presents the development of a dome-shaped tactile finger (Barodome) capable of detecting decoupled normal and shear forces, as well as the contact location, using finite embedded barometers and a trained neural network.

Strength: Tactile sensing is emerging in the field of robotics. Understanding how sensor morphology affects its performance and characteristics is crucial for sensor design and development. Therefore, the research problem addressed in this paper is significant, and the contribution of the paper may facilitate optimal sensor design through theory-informed analysis.

Weakness: There is still room for improvement and further elaboration in the paper. In fact, the justification of the tactile superresolution theory for taxel-based tactile sensors has been well demonstrated in the authors' previous work [23]. While the paper introduces an improvement in the model to account for the effect of shear forces, it is crucial to clarify how this enhancement significantly contributes to the sensor design process. Additionally, it is essential to experimentally verify whether the model can indeed guide a better version of a tactile sensor compared to others. In its current form, the theory and the practical implementation (Barodome sensor design) are somewhat loosely connected, although the practical sensor implementation may validate some implications from the theory-based analysis. Below are detailed comments and suggestions for improvement.

Comments:

1. It may be worth adding additional related works on tactile sensors with 3D force detection capability. These studies could serve as benchmarks to verify the improvements of the proposed Barodome design over the state of the art.
2. In addition to predicting system errors caused by shear forces, it would be more interesting to see how the model/theory informs different sensing performance or accuracy by varying the arrangement of taxels. For example, factors like the distance between taxels, the number of taxels, and their grid patterns could be investigated.
3. Can the model generalize to sensors with various form factors (e.g., different shapes and sizes)?
4. Can the model predict sensing performance with varying material properties of the soft membrane, such as different Shore hardness levels?
5. Is it possible to predict the influence of multi-point contact, and how effective is the tactile superresolution for multi-point contact scenarios?

In summary, in the current version of the paper, the practical implementation has not clearly demonstrated whether the model effectively informs optimal (better) performance for taxel-based tactile sensor design. Additionally, it remains unclear whether the model and analysis can lead to a desirable sensor design. At the very least, there should be a discussion or justification on how desirable sensor design can be achieved based on the proposed model and theory.

(Remarks on code availability)

Reviewer #2

(Remarks to the Author)

Summary of Paper

This paper presents a theory for understanding superresolution on a tactile array or dome. It builds on previous work on tactile isolines (23) to account for shear forces and how applied shear forces affect the uncertainty in position estimations. The paper also introduces a new sensor, Barodome, whose design is guided by the isoline theory and data. The authors compare performance of Barodome using a learned model with predicted performance with and without the presence of shear forces. The primary contributions are 1) the tactile isoline model in the presence of shear and 2) the design and characterization of Barodome.

Major Comments

The paper provides very thorough analysis and presentation of the isoline theory in the presence of shear, particularly figures 2, 3, and 4. I really like all of the visualizations here! They clearly show how isolines change depending on sensor placement, how shear forces shift the isolines, how isolines from different sensors intersect when multiple sensors are activated, etc.

The methods and design are described thoroughly and completely. They could be replicated by someone else.

The Barodome is a nice application of the isoline theory. However, the raw signals (either of the barodome or the F/T sensor) are not shown in any figures. I would find this helpful (perhaps as an inset to one of the figures (Fig. 5?)) in order to get a sense for the SNR, hysteresis, and what signals the learned model is seeing.

Minor Comments

Line 39, all of the resistive citations are self-citations. There are many other resistive sensor works that could be cited in addition here. (E.g. touchlab.io, <https://doi.org/10.1038/s41586-019-1234-z>, and the references in Fig. 1 of <https://doi.org/10.1016/j.robot.2015.07.015>, among others)

Line 143 talks about how a floating sensor enhanced force sensitivity. Do you have any idea of why this is the case?

Line 161, can you support this claim by citing other work that differentiates object size but does not study the mechanism behind it?

Line 166: Can you cite an example of a sensor using specific directions to decouple normal and shear forces

I'm confused by what Fig. 3e(i) is showing, could you improve the caption for this?

I'm confused by what Fig. 6c(i) is showing, specifically what each shaded region shows and why the forces are discretized here. Could you improve the description of this?

Just a thought / discussion point: Have the authors considered how torsional shear would affect isolines? This could be an interesting discussion point

Just a thought / discussion point: The paragraph starting on line 151 discusses how smaller indenters require less force to activate the sensor to the same level. It could be interesting to (additionally) plot Figure 2c as surface pressure (i.e. force / contact area) vs lateral position. From the discussion in this paragraph, I'm unsure whether you'd expect isolines of pressure vs position to be the same across different indenters or not. Along these lines, I'm also curious if you'd see better force performance across different indenters (Fig 6d) if you were to estimate both surface pressure and contact area and infer the applied force from that. I am thinking about this because in figure 6d the model estimates force across different indenter shapes, but we also know that different indenters have very different isolines (Fig 2c). If the surface pressure vs position isolines are more similar across different indenters, perhaps the model would be able to estimate this more easily?

Typos, grammatical errors, etc

Line 139: "constraint" to "constrained"

Line 150: "sizes" to "diameters"

Line 182: "a deeper insights" to "a deeper insight" or "deeper insights"

Line 209: "interact" to "interacts"

Line 245: "extend" to "extent"

Fig. 1a and Fig. 5a have some overlapping text

Fig. 6a(ii): "Radical" to "Radial"

(Remarks on code availability)

The manuscript says that data and code will be released upon acceptance

Reviewer #3

(Remarks to the Author)

Comments:

The paper has been written well, with good supporting data. Additional explanations are required to make the paper easier to understand for the readers.

1. Although the article mentions that there are some limitations in current tactile sensors, it does not fully demonstrate the advantages of this method over existing methods. It is recommended to compare your method with other models in a table.
2. The contact objects used in the experiment (such as several spherical indenters and a cylindrical indenter) are relatively simple and may not fully represent the diversity of complex contact objects in reality. Has the author considered the influence of objects with more complex shapes and materials on the sensor? Will these influences significantly affect the performance of the sensor?
3. The article labels the sensor as having superresolution, but is the accuracy of 0.8mm in localization positioning accuracy of the sensor excellent enough? Can it be compared with other tactile sensors of the same type in terms of parameters?
4. What is the influence of noise on the sensor? How is the measurement noise dealt with?
5. For the part of sensor performance evaluation, has the long - term use stability and reliability of the sensor been considered?

(Remarks on code availability)

Version 1:

Reviewer comments:

Reviewer #1

(Remarks to the Author)

The manuscript has been polished with more detailed explanations and revisions. However, the following points should be clarified before publication.

The revised manuscript states: "Analysis of a single sensor unit enables the precise adjustment of hyperparameters in our theoretical model (Eq. 5), ensuring accurate predictions of optimal performance for fully integrated sensing devices of any shape or size. Through a comparison of desired performance goals with the predictions, we can effectively refine the structural and material aspects to achieve our objectives before physically building a whole sensor." Since the paper presents a practical implementation for only one sensor configuration, it is difficult to claim that the model can ensure optimal performance for devices of any shape and size without substantial experimental validation.

Additionally, in the Theory-Informed Tactile Sensor Design section, the statement "These design choices were informed by the theoretical analysis mentioned above" would be clearer if it explicitly specified the design choices and theoretically optimal performance indicated by the model for guiding the design of the current Barodome. The extent to which the performance and accuracy of Barodome align with the model's predictions should be thoroughly discussed, as this is crucial for assessing the validity of the model in 3D-shaped sensor design.

Finally, it would be great to include a video demonstration showcasing Barodome's real-time operation.

(Remarks on code availability)

The code may be difficult to reproduce without detailed instructions on usage and hardware implementation.

Reviewer #2

(Remarks to the Author)

I am reviewing this paper for the second time. In the first round, I was reviewer #2. The authors have thoroughly addressed many of the comments from all 3 reviewers, including adding a table comparing the sensor and superresolution results to other works, elaborating and better explaining many sections, adding raw signal plots, adding addition plots to figure 2, and adding a number of additional relevant references. I feel these changes have adequately addressed the comments.

My only remaining comment is that in the added plots in figure 2d, the rebuttal document says that these show "surface pressure versus lateral positions", however the actual figure 2d is labeled as force vs position. That said, if these are force vs position plots, then some of the curves do not match up with fig. 2c in the way that I expected. For example, the dark blue

(10kPa, D=20) curve in 2d does not visually match the dark blue (D6, 10kPa) curve in 2c. Likewise, the dark blue (0.5kPa, D=20) curve in 2d does not visually match the red (D6, 0.5kPa) curve in 2c. So, either the label or the data seems to be wrong here.

(Remarks on code availability)

Reviewer #3

(Remarks to the Author)

Although current superresolution-based sensors (Barodome) are limited to resultant force and struggle to analyze local force distribution information on surfaces of complex-shaped objects, it is still necessary to experimentally validate their perception boundaries for such objects. It is recommended to supplement the "Results" section with qualitative test results for these objects (e.g., cubes or prisms), such as inferring contact regions through superposition of multi-indenter response models. Even if such experiments are preliminary explorations, they could demonstrate the authors' systematic delineation of the sensor's applicability boundaries and provide theoretical guidance for future improvements (e.g., integrating dense taxel layouts).

(Remarks on code availability)

Version 2:

Reviewer comments:

Reviewer #1

(Remarks to the Author)

The manuscript has been substantially improved during the second round of revision by clarifying its contributions and significance. I therefore recommend the paper for publication.

(Remarks on code availability)

The provided code and data are likely sufficient to support the reproduction of the results. However, additional usage instructions may be needed to ensure that others can effectively use this resource to replicate the work and advance further research.

Reviewer #2

(Remarks to the Author)

My concerns have been addressed and I don't have any additional comments. Thank you for fixing the bug in figure 2 and for posting your code.

(Remarks on code availability)

The code provides a jupyter notebook for training models and reproducing figures. The code and data are linked in the manuscript.

Reviewer #3

(Remarks to the Author)

The authors have accurately addressed my concerns and made sufficient revisions in the manuscript. I support the normal acceptance of this paper.

(Remarks on code availability)

Answers to Reviewer 1

Thank you for your comments, especially your valid concern regarding the introduction of benchmarks. We have added new experiments and analyses to verify the improvements of our proposed Barodome design over state-of-the-art designs.

While the paper introduces an improvement in the model to account for the effect of shear forces, it is crucial to clarify how this enhancement significantly contributes to the sensor design process. Additionally, it is essential to experimentally verify whether the model can indeed guide a better version of a tactile sensor compared to others. In its current form, the theory and the practical implementation (Barodome sensor design) are somewhat loosely connected, although the practical sensor implementation may validate some implications from the theory-based analysis.

We conducted new experiments to further validate the performance of our Barodome design. Our theory-informed Barodome achieves over 1200 times superresolution under normal forces and 120 times superresolution in contact localization accuracy under shear forces, outperforming all other sensors listed in Table 1. Only a few tactile sensors have been reported to accurately predict contact locations under shear forces.

Furthermore, we emphasize that the key contribution of our established theory is to provide a clear and reasonable explanation for the system-level inherent accuracy loss caused by shear forces compared to normal forces. This contribution is theoretical rather than purely methodological. The observed accuracy loss under shear forces is a well-recognized challenge in the tactile sensing community but has remained inadequately explained. Our paper addresses this gap by offering a robust theoretical foundation to better understand and quantify this phenomenon.

Building on our prior work [27], this paper further refines the design of Barodome, with many design decisions guided by the analysis in that earlier study. Key considerations include taxel spacing, taxel count, grid patterns, sensor shapes and sizes (evolving from 2D in [27] to 3D in this paper), material properties of the soft membrane (such as Young's modulus and Poisson's ratio), and the influence of multi-point contact.

These factors were comprehensively analyzed in our previous work [27], providing a strong foundation for the current study. In this manuscript, we focus more on the interaction between sensors and objects. Thus, we extend our analysis to include additional aspects, such as the structural properties of the sensor elastomer, the sizes of indenters, and the effects of force directions. By jointly considering these factors, we present a comprehensive approach that allows for a deeper understanding and optimization of Barodome's performance.

Additionally, we've modified our Conclusion section text to make further clarification: "Analysis of a single sensor unit enables the precise adjustment of hyperparameters in our theoretical model (Eq. 5), ensuring accurate predictions of optimal performance for fully integrated sensing devices of any shape or size. Through a comparison of desired performance goals with the predictions, we can effectively refine the structural and material aspects to achieve our objectives before physically building a whole sensor."

1. It may be worth adding additional related works on tactile sensors with 3D force detection capability. These studies could serve as benchmarks to verify the improvements of the proposed Barodome design over the state of the art.

Thank you for your suggestion. We have added ten additional references to more comprehensively reflect the breadth of research in the field of tactile sensors with 3D force detection capabilities. These references help to position our work in the context of existing benchmarks and provide a more thorough comparison of the advancements made by our Barodome design over the state-of-the-art sensors. We have also included a new table (Table 1) in the main text to summarize state-of-the-art 2D and 3D tactile sensors with super-resolution capabilities for detecting contacts. As shown in Table 1, our sensor, Barodome, achieves contact location accuracies of 0.17 mm under normal forces and 0.52 mm under shear forces when the indentation spacing is 1 mm. Notably, only a few tactile sensors have been reported to accurately predict contact locations under shear forces, and Barodome outperforms them.

2. In addition to predicting system errors caused by shear forces, it would be more interesting to see how

the model/theory informs different sensing performance or accuracy by varying the arrangement of taxels. For example, factors like the distance between taxels, the number of taxels, and their grid patterns could be investigated.

We agree with the reviewer’s valid concern that the distance between taxels, the number of taxels, and their grid patterns are crucial factors. These elements were thoroughly addressed and implemented in our previous paper [27]. To provide clarity here, we have included relevant figure from [27] below for reference.

Distance between taxels: In [27], we developed two types of one-dimensional sensors. One type, shown in [27] Figure 3E, consists of six barometers with a spacing of approximately 6.5 mm between each sensor. The other type, shown in [27] Figure 3F, features six barometers with varying but generally larger spacing distances. The localization accuracies illustrated in [27] Figure 3E(i) and 3F(ii) clearly demonstrate that a closer placement of taxels enhances both sensitivity and accuracy.

Fig. 3. Superresolution in 1D. Real line sensor device with six barometer taxels at a distance of about 6.5 mm. (A) Sensor device with stimulation testbed. (B) Taxel response for different indentation depths (i) and resulting TVIs for different sensor values (ii). (C) Analytical calculation based on the theory performed for one of the taxels. In (i), we assume that the sensor measurement noise is constant with $\sigma_s = 20.0$ Pa. The position and force SD (v) are analytically derived from the approximated TVIs (ii) with appropriate parameters c, α, λ (iii, iv). (D and E) Quantitative evaluation of the sensor device using ML models to infer the position and force magnitude. [D(i)] SR factor depending on the contact force magnitude as predicted by the theory without and with testbed precision, and as achieved by the ML solution and the numeric solution. [D(ii)] Position and force error of the ML solution depending on stimulation force: mean and SD over the 32.5-mm sensing surface between taxel 1 and 6. (E) Spatially resolved position error (i) and force error (ii). The orange lines are the TVIs for the smallest sensor value [500 Pa in B(ii)]. (F) TVIs (i) and spatially resolved position error (ii) for another 1D sensor layout with varied adjacent sensor distances.

Number of taxels: In [27], we provide both theoretical modeling (Figure 2C(i)) and experimental results (Figure 3E and F) to demonstrate the influence of the number of taxels on sensor performance. Our super-resolution theoretical model informs that:

“In the area where more than two taxels respond to the contact stimulation, higher localization accuracy is possible, as shown in Fig. 2C(i). This leads to a reduction in uncertainty about the contact force due to two effects. The first one is the averaging of independent noise sources leading to a factor of $\sqrt{(1/a)}$, where a is the number of active taxels. The second effect comes from the intersection of TVIs from more distant taxels, which, due to a higher TVI slope, results in a lower uncertainty as apparent from Eq. 7. The sensitivity F_S is shown in Fig. 2C(ii) as a dash-dotted lower bound, together with the accuracy of localization σ_P and force quantification σ_F . The sensitivity is not homogeneous and is higher between taxels (lower minimum force), which might be an unexpected result at first glance. In summary, the most important take-home message is to have multiple taxels responding to a contact force because it improves accuracy.”

Fig. 2. Theory of superresolution sensing in 1D. (A) Model used for contact point localization at SR (Ω ; Eq. 4). [A(i)] shows the intersection of TVIs of two sensors (marked as S-1 and S-2). The place where the lines corresponding to the particular sensor readings cross is the contact location that is marked as p_1 . The measurement noise σ_5 leads to uncertainty of position σ_p and force σ_f . [A(ii)] Different intersection types. (B) Effect of TVI shape on SR characteristics. [B(i)] TVIs are modeled as $|d|^\alpha + \text{const}$ for two taxels (at distance 1) with different attenuation exponents α . Note the different shapes of the intersection areas. [B(ii)] Resulting sensitivity (minimum force), SD of position localization, and SD of force inference. (C) Theoretical SR characteristics of a 1D sensor with multiple taxels. [C(i)] Three taxels localizing a single contact. Uncertainty is decreased because of averaging of independent noise sources and larger TVI slopes. [C(ii)] The spatial distribution of accuracy for a single contact. Below the orange dash-dotted line (sensitivity), no SR localization is possible. Notice the increase in accuracy for higher forces because multiple taxels are activated.

Grid Patterns In [27], we compute the sensitivity distribution across the sensing surface for two sensor layouts: a classic grid arrangement and a honeycomb layout. The honeycomb pattern results in a more homogeneous sensitivity distribution, as illustrated in Fig. S7C.

Fig. S7. Taxel values isolines for a 2D sensing surface. **A(i):** Taxel value isoline (iso-surface) for a single taxel. **A(ii, iii):** Intersection volume (due to measurement uncertainty) for two taxels at a distance D along the x axis, see top view. The localization would be very uncertain along the y direction. **B:** Intersection volume for different attenuation exponents α . **C:** Proper localization requires at least 3 taxels for 1 contact point. **(i)** shows a hexagonal sensor placement with resulting sensitivity over the surface and an illustration of the intersection volume of the TVIs from 3 taxels. **(ii)** is the same for a grid and 4 taxels. **(iii)** illustrates a spurious contact localization for 2 contact points and 6 taxels (marked with “?”), however its uncertainty would be very large. **(iv)** illustrates the intersection of TVIs for the case of a curved sensing surface. The same applies as in the case of planar 2D, except that distances need to be measured as geodesics on the curved surface.

3. Can the model generalize to sensors with various form factors (e.g., different shapes and sizes)?

Our model first focuses on analyzing the behavior of a single sensing unit (taxel), considering factors such as structure, material properties, electrical signal noise, contact object shape, interaction force amplitude and direction, as well as multi-point contact. This analysis is generalizable to sensors of various shapes and sizes, as they can be viewed as compositions or arrays of multiple sensing units. For example, our Barodome features a 3D fingertip-shaped structure, which differs from the 2D sensor introduced in our previous work [27]. Notably, the Barodome achieves a superresolution factor of 1200 under normal forces,

similar to the performance achieved by the 2D sensor in our previous work.

We have added a new table (Table 1) in the current manuscript for clearer presentation. To provide further clarity, we have also included additional text in the Conclusion section:

“Analysis of a single sensor unit enables the precise adjustment of hyperparameters in our theoretical model (Eq. 5), ensuring accurate predictions of optimal performance for fully integrated sensing devices of any shape or size. Through a comparison of desired performance goals with the predictions, we can effectively refine the structural and material aspects to achieve our objectives before physically building a whole sensor.”

4. Can the model predict sensing performance with varying material properties of the soft membrane, such as different Shore hardness levels?

Yes, our model also incorporates an analysis of material properties and their influence on sensing performance, focusing on two key aspects: Young’s modulus (stiffness or softness) and Poisson’s ratio. Young’s modulus quantifies how easily a material deforms under stress and has a direct, proportional effect on deformation. A soft material (i.e., one with a low Young’s modulus) can enhance sensitivity but may also be more susceptible to inertial effects, especially if the material has a high density. This susceptibility can compromise performance under dynamic conditions or when subjected to strong interaction forces. Poisson’s ratio measures the relative transverse expansion of a material when compressed axially. Most elastomers have a Poisson’s ratio close to 0.5, indicating near-incompressible behavior, while metals typically have a Poisson’s ratio around 0.3.

Additionally, we would like to clarify a potential misunderstanding regarding **hardness**. Unlike Young’s modulus, which pertains to elastic deformation, hardness describes a material’s resistance to surface deformation, such as scratching, indentation, or abrasion. It primarily refers to plastic deformation, not elastic behavior. This distinction is crucial when evaluating materials for their role in sensing applications. Nowadays, the Shore hardness scale is commonly used to quantify how easily a soft material deforms. Softer materials typically have a lower Young’s Modulus (which we use in our analysis) and correspondingly lower hardness values.

The detailed analysis was introduced in our previous research [27]. To provide clarity, we have included the relevant figure from [27] below for reference:

Fig. S3. Physical Factors Influencing TVIs using FEM. **A(i):** The model: because of symmetry, only one half is shown. The thickness is $T = 10$ mm, and we consider taxels at level $L1 = T - 1$ mm, $L2 = T - 5$ mm and $L3 = T - 9$ mm. **A(ii):** Total displacement maps as well as its x and z components. **B:** TVIs for hypothetical sensors measuring total displacement or component-wise displacement, for different taxel depth **(i)**, material thickness **(ii)**, Poisson’s ratios **(iii)**, and indenter shapes **(iv)**. TVIs for negative sensor values are in a light shade. Default configurations are: Poisson’s ratio $\nu = 0.49$, Young’s modulus $E = 0.07$ MPa, and density $\rho = 1.07$ g/cm³ for the transmission medium; Poisson’s ratio $\nu = 0.33$, Young’s modulus $E = 71$ GPa, and density $\rho = 2.77$ g/cm³ for the indenter; bonded contact type without friction consideration.

To aid in comprehending the Conclusion section, we have added relevant discussion text for better elaboration and context.

“As concluded in [27], the key properties of the soft transmission medium are Young’s modulus and Poisson’s ratio. A lower Young’s modulus, with similar TVIs, enhances sensitivity but increases deformation under inertia and strong forces. A lower Poisson’s ratio (0.5 for elastomers, 0.3 for metals) alters TVIs, reducing radial sensitivity while increasing depth sensitivity.”

5. Is it possible to predict the influence of multi-point contact, and how effective is the tactile superresolution for multi-point contact scenarios?

Our model can predict the influence of multi-point contact and demonstrates the conditions necessary for accurate inference in such scenarios. The essential analysis was introduced in our previous work [27]. To provide clarity, we have included the relevant figure from [27] below for reference:

Fig. S2. Multiple contacts discrimination. There are six sensing units aligning in one line forming a 1D layout to discriminate double contacts. When one contact locates in the left green-shaded area, all contact situations of the second contact can be successfully discriminated when they are in the right green-shaded area. In contrast, when the second point is too close, say in the gray-shaded area, the following problem occurs. The first contact at p_1 alone would activate two sensing units (S-2: orange-dotted line, S-3: gray-dotted line). Another contact in the gray-shaded area at p_2 would activate three sensing units (S-3: gray-dashed line, S-4: orange-dashed line, S-5: orange-dashed line). When these two contacts happening simultaneously, the value of the sensing unit S-3 turns to be an orange-dash-dotted line (combined TVI). This will result in an unreliable inference. Multiple spurious contact points occur ("★") rather than two actual contact points ("●")

To address the effectiveness of tactile superresolution for multi-point contact scenarios, we have expanded the discussion in the manuscript (Conclusion section), focusing on the influence of complex contact objects: “Complex contact objects can be modeled as combinations of indenters with varying diameters. According to [27], distinguishing multiple contacts requires at least one taxel’s distance between them, with at most one taxel activated per contact. Higher contact forces increase the minimum distinguishable distance for double contacts. Spatially continuous contact objects needing detailed force distribution prefer dense taxel layouts, such as arrays or vision-based tactile sensors, as current superresolution-based sensors, limited to resultant force, are inadequate. Additionally, the deformation and contact diameter variations of soft objects during interaction require further study.”

In summary, in the current version of the paper, the practical implementation has not clearly demonstrated whether the model effectively informs optimal (better) performance for taxel-based tactile sensor design. Additionally, it remains unclear whether the model and analysis can lead to a desirable sensor design. At the very least, there should be a discussion or justification on how desirable sensor design can be achieved based on the proposed model and theory.

This is a valid and insightful point. We have used your comments to thoroughly revise and enhance our manuscript, addressing them through detailed, point-by-point answers as outlined above. We look forward to your feedback on the updated version of our manuscript.

Thank you for your thoughtful review and valuable input.

Answers to Reviewer 2

We are grateful for your constructive feedback, especially regarding aspects like torsional forces and the plot of surface pressure versus lateral position for better elaboration and understanding. We find these suggestions very helpful and have incorporated or modified the related content in our manuscript accordingly.

The Barodome is a nice application of the isoline theory. However, the raw signals (either of the barodome or the F/T sensor) are not shown in any figures. I would find this helpful (perhaps as an inset to one of the figures (Fig. 5?)) in order to get a sense for the SNR, hysteresis, and what signals the learned model is seeing.

Useful comments! We have added the noise levels of a barometer reading and the force/torque sensor value in the z direction (normal force) under no-contact conditions in Figure 5d. The response of the barometer reading with increasing normal force under contact is also illustrated. Regarding the signal-to-noise ratio (SNR), the noise level has a variance of 15 Pa over a measurement range of 90,000 Pa. Our sensor is characterized only in static and quasi-static cases. Currently, evaluations do not consider material hysteresis, which we plan to address in future work. The machine learning model (MLP) uses sixteen barometer readings as input to predict directional position and force values.

Line 39, all of the resistive citations are self-citations. There are many other resistive sensor works that could be cited in addition here. (E.g. touchlab.io, <https://doi.org/10.1038/s41586-019-1234-z>, and the references in Fig. 1 of <https://doi.org/10.1016/j.robot.2015.07.015>, among others)

We appreciate your suggested references. In response, we have added additional references to more accurately reflect the breadth of research in this field:

[9] S. Teshigawara, S. Shimizu, K. Tadakuma, M. Aiguo, M. Shimojo, and M. Ishikawa, "High sensitivity slip sensor using pressure conductive rubber," in *Proceedings of the IEEE SENSORS*, 2009, pp. 988–991.

[11] S. Sundaram, P. Kellnhofer, Y. Li, J.-Y. Zhu, A. Torralba, and W. Matusik, "Learning the signatures of the human grasp using a scalable tactile glove," *Nature*, vol. 569, no. 7758, pp. 698–702, 2019.

[13] Z. Kappassov, J.-A. Corrales, and V. Perdereau, "Tactile sensing in dexterous robot hands Review," *Robotics and Autonomous Systems*, vol. 74, pp. 195–220, 2015.

[14] M. Chen, W. Luo, Z. Xu, X. Zhang, B. Xie, G. Wang, and M. Han, "An ultrahigh resolution pressure sensor based on percolative metal nanoparticle arrays," *Nature Communications*, vol. 10, no. 1, p. 4024, Sep 2019.

[15] T. Taunyazov, W. Sng, H. H. See, B. Lim, J. Kuan, A. F. Ansari, B. C. K. Tee, and H. Soh, "Event-Driven Visual-Tactile Sensing and Learning for Robots," in *Proceedings of Robotics: Science and Systems (RSS)*, Corvallis, Oregon, USA, July 2020.

Line 143 talks about how a floating sensor enhanced force sensitivity. Do you have any idea of why this is the case?

When probed by an indenter, the soft elastomer deforms the most near the surface and the least near the fixed boundary. This deformation leads to a local pressure increase, with the pressure being higher near the surface due to the localized pressure increase towards the surface direction. We added an elaboration stating, "due to a local pressure increase towards the surface direction," to clarify that the deformation causes a localized pressure increase, with the pressure being higher near the surface.

Line 161, can you support this claim by citing other work that differentiates object size but does not study the mechanism behind it?

[34] G. Li, S. Liu, L. Wang, and R. Zhu, "Skin-inspired quadruple tactile sensors integrated on a robot hand enable object recognition," *Science Robotics*, vol. 5, no. 49, p. eabc8134, 2020.

This work demonstrates the ability to differentiate object sizes but does not explore the underlying mechanism responsible for this capability.

Line 166: Can you cite an example of a sensor using specific directions to decouple normal and shear forces

[22] Y. Yan, Z. Hu, Z. Yang, W. Yuan, C. Song, J. Pan, and Y. Shen, “Soft magnetic skin for super-resolution tactile sensing with force self-decoupling,” *Science Robotics*, vol. 6, no. 51, 2021.

[29] T. Paulino, P. Ribeiro, M. Neto, S. Cardoso, A. Schmitz, J. Santos-Victor, A. Bernardino, and L. Jammone, “Low-cost 3-axis soft tactile sensors for the human-friendly robot Vizzy,” in *2017 IEEE International Conference on Robotics and Automation (ICRA)*, 2017, pp. 966–971.

I’m confused by what Fig. 3e(i) is showing, could you improve the caption for this?

We modified Figure 3e(i) accordingly.

I’m confused by what Fig. 6c(i) is showing, specifically what each shaded region shows and why the forces are discretized here. Could you improve the description of this?

We modified the caption for better understanding, “(c) compares the sensing performance of pure normal forces with coupled shear forces in inferring contact location (left) and force strength (right). Errors are evaluated based on resultant force intervals for both “Normal” (pure normal forces, shown in purple) and “Total” (combined forces including shear components, shown in orange) [c(i)]. For each “Total” force interval, directional position accuracies are analyzed separately for X (red), Y (green), Z (blue), allowing for comparative assessment.”

Just a thought / discussion point: Have the authors considered how torsional shear would affect isolines? This could be an interesting discussion point

We have added a discussion in the Conclusion section: “Torsional shear forces, similar to tangential shear forces, create a pull-away effect, reduce sensor readings, and necessitate higher normal forces. It is important to note that torsional shear forces require an offset contact distance to influence the sensor unit effectively.”

Just a thought / discussion point: The paragraph starting on line 151 discusses how smaller indenters require less force to activate the sensor to the same level. It could be interesting to (additionally) plot Figure 2c as surface pressure (i.e. force / contact area) vs lateral position. From the discussion in this paragraph, I’m unsure whether you’d expect isolines of pressure vs position to be the same across different indenters or not. Along these lines, I’m also curious if you’d see better force performance across different indenters (Fig 6d) if you were to estimate both surface pressure and contact area and infer the applied force from that. I am thinking about this because in figure 6d the model estimates force across different indenter shapes, but we also know that different indenters have very different isolines (Fig 2c). If the surface pressure vs position isolines are more similar across different indenters, perhaps the model would be able to estimate this more easily?

This is indeed a great idea. As suggested, we have added a subplot to Figure 2, now labeled as Figure 2d, showing surface pressure versus lateral positions with respect to indenters of different sizes. As illustrated in Figure 2d, smaller indenters require less force to activate the sensor to the same pressure level, as they induce a more localized pressure increase in the sensor readings.

We elaborated on this in the main text: “Based on the analysis in [27], smaller $\alpha(s)$ values approaching two, along with larger perception fields that activate multiple sensors, result in higher accuracy. Notably, larger indenters exhibit both of these characteristics.”

Typos, grammatical errors, etc Line 139: “constraint” to “constrained” Line 150: “sizes” to “diameters” Line 182: “a deeper insights” to “a deeper insight” or “deeper insights” Line 209: “interact” to “interacts” Line 245: “extend” to “extent” Fig. 1a and Fig. 5a have some overlapping text Fig. 6a(ii): “Radical” to “Radial”

Thank you for pointing out all these typos; we have made the necessary corrections.

The manuscript says that data and code will be released upon acceptance

The data size is approximately 500 MB, and we are in the process of setting up permanent storage to upload the data. One temporary solution is to share the data upon email request. The current code is available at code.

Answers to Reviewer 3

Thank you very much for your comments. We realized that we had inadvertently omitted several details in our initial submission, which made it less straightforward for readers to understand. We have now added additional text and experimental results to better elaborate on our findings and conclusions.

1. Although the article mentions that there are some limitations in current tactile sensors, it does not fully demonstrate the advantages of this method over existing methods. It is recommended to compare your method with other models in a table.

This is a good suggestion. we have included a new table (Table 1) in the main text to summarize state-of-the-art 2D and 3D tactile sensors with superresolution capabilities for detecting contacts. As shown in Table 1, our sensor, Barodome, achieves contact localization accuracies of 0.17 mm under normal forces and 0.52 mm under shear forces when the indentation spacing is 1 mm. Our theory-informed design Barodome achieves over 1200 times superresolution under normal forces and 120 times superresolution in contact localization accuracy under shear forces, outperforming all other sensors listed in Table 1. Only a few tactile sensors have been reported to accurately predict contact locations under shear forces.

Furthermore, we aim to emphasize that the key contribution of our established theory is to provide a reasonable explanation for the system-level inherent accuracy loss caused by shear forces compared to normal forces. This is a theoretical explanation, rather than a purely methodological approach. This issue represents a broader challenge within the tactile sensing community, where such inherent accuracy loss under shear forces is widely observed but remains inadequately explained. Our paper seeks to fill this gap by offering a clear theoretical foundation for it.

2. The contact objects used in the experiment (such as several spherical indenters and a cylindrical indenter) are relatively simple and may not fully represent the diversity of complex contact objects in reality. Has the author considered the influence of objects with more complex shapes and materials on the sensor? Will these influences significantly affect the performance of the sensor?

We added following text to address your concern: “Complex contact objects can be modeled as combinations of indenters with varying diameters. According to [27], distinguishing multiple contacts requires at least one taxel’s distance between them, with at most one taxel activated per contact. Higher contact forces increase the minimum distinguishable distance for double contacts. Spatially continuous contact objects needing detailed force distribution prefer dense taxel layouts, such as arrays or vision-based tactile sensors, as current superresolution-based sensors, limited to resultant force, are inadequate. Additionally, the deformation and contact diameter variations of soft objects during interaction require further study.”

3. The article labels the sensor as having superresolution, but is the accuracy of 0.8mm in localization positioning accuracy of the sensor excellent enough? Can it be compared with other tactile sensors of the same type in terms of parameters?

To address your raised concern, we have run new experiments, added new results, texts and a table (Table 1) to our manuscript.

The localization accuracy of 0.8 mm is achieved when the indentation spacing is 1 mm, and the evaluation data is entirely separate from the training dataset, containing no previously seen positions. The machine learning model is required to generalize effectively within a 2 mm range. If the indentation spacing is narrower, as in our previous work [27], where the sensor had an indentation spacing of 0.5 mm, it achieved a localization accuracy of 0.16 mm. In contrast, if the evaluation data includes positions previously seen during training but not the same force levels, the machine learning model achieves a much higher localization accuracy of 0.17 mm. For both cases, the observed loss of inherent accuracy at the system level of 0.35 mm ($0.35 = 0.52 - 0.17$) and 0.5 mm ($0.5 = 1.3 - 0.8$) closely matches the theoretical prediction of 0.33 mm. In Table 1, we compare our Barodome sensor with other state-of-the-art sensors using similar evaluation metrics, and our Barodome sensor demonstrates superior performance.

New text is “As higher forces are applied, localization and force quantification improve, achieving 0.8 mm and 8% for normal forces, and 1.3 mm and 15% for coupled shear forces, with reduced median error and

narrower median absolute deviation. When the evaluation data is not part of the training dataset but includes positions seen in the training dataset, the position and force accuracies (root-mean-square error, RMSE) can reach up to 0.17 mm and 1% for normal forces, and 0.52 mm and 4% for coupled shear forces.”

4. What is the influence of noise on the sensor? How is the measurement noise dealt with?

Thank you for raising this important question. We realized that we had inadvertently omitted these details in our initial submission. We have now revised the manuscript to include thorough information and have incorporated it into the main text and Figure 5d to address this concern comprehensively.

“The sensor operates on a machine-learning-driven system, utilizing a trained model to predict contact location (P_x, P_y, P_z) and directional force magnitudes (F_N, F_{S1}, F_{S2}) for a single contact based on raw measurements of barometers (B_1, \dots, B_{16}). The barometric sensing unit, with a full measurement range of 90,000 Pa, exhibits a noise level characterized by a variance of 15 Pa, as illustrated in Figure 5d. To conduct the experiments, we developed a specialized testbed (Figure 5c(i)) with 3 degrees of freedom (DoF) for translation (Barch Motion Linear Guide has a position resolution of 0.0075 mm) and 2 DoF for rotation (Dynamixel MX-28AT and MX-64 AT result in a maximum position error of 0.03 mm, as shown in Figure 6a(ii)). This setup carries a four-millimeter spherical indenter to probe the **Barodome** sensing surface with both normal and shear forces applied. A force sensor (ATI-Mini 40 with a measurement noise level in variance of 4.0 mN) is attached between the probe and the testbed to quantify the contact force magnitudes (Figure 5c(ii)). A multilayer-perceptron (MLP) structured machine-learning model is trained with a least-square cost function on the collected data that maps raw barometer values to contact information during runtime (Figure 5d). Model uncertainties as Gaussian residuals stem from barometric sensor noise, testbed accuracy limitations, force sensor noise, and data size constraints. By leveraging the equivalence between minimizing model uncertainty through maximum likelihood estimation and minimizing least-squares error for Gaussian residuals (attributed to barometric sensor noise), the MLP is directly optimized to predict the quantities of interest, as shown in Figure 1c(ii).”

5. For the part of sensor performance evaluation, has the long - term use stability and reliability of the sensor been considered?

We observed a slight drift in localization and force quantification over time. We have added the following text to elaborate on our sensor’s performance regarding long-term use stability and reliability.

“After approximately 300,000 intensive contact cycles over two weeks, including **both normal and shear forces**, the Barodome exhibited position accuracy offsets of ($P_x = -0.02 \text{ mm}, P_y = 0.04 \text{ mm}, P_z = 0.12 \text{ mm}$) and force accuracy offsets of ($F_x = -0.03 \text{ N}, F_y = 0.05 \text{ N}, F_z = 0.15 \text{ N}$) for the 20 mm spherical indenter, presumably due to a shift in the sensing units. Further endurance testing is necessary to assess the long-term stability and reliability of the system for various applications, which will be addressed in future work. Future work will focus on mitigating this drift through material improvements and advanced calibration techniques.”

May 23, 2025

Manuscript Tracking Number: NCOMMS-24-41761A

Manuscript Title: Sensing multi-directional forces at superresolution using taxel value isoline theory

Authors: *Huanbo Sun, Adam Spiers, Hyosang Lee, Jonathan Fiene, and Georg Martius*

Answers to Reviewer 1

Thank you for your valuable comments. We have used them to improve the manuscript by adding clarifying text, particularly a step-by-step explanation of how we apply the theory to guide sensor design, predict inherent performance loss, and validate the model through real-world experiments.

The revised manuscript states: “Analysis of a single sensor unit enables the precise adjustment of hyper-parameters in our theoretical model (Eq. 5), ensuring accurate predictions of optimal performance for fully integrated sensing devices of any shape or size. Through a comparison of desired performance goals with the predictions, we can effectively refine the structural and material aspects to achieve our objectives before physically building a whole sensor.” Since the paper presents a practical implementation for only one sensor configuration, it is difficult to claim that the model can ensure optimal performance for devices of any shape and size without substantial experimental validation.

Thank you for pointing out this valid concern. We acknowledge that our original wording may have overstated our contribution, as the theory in this manuscript is validated using only a single sensor configuration. To clarify and ensure our claims are technically rigorous, self-contained, and appropriately justified within the scope of the current work, we have revised the statement as follows: “*ensuring accurate predictions of optimal performance for fully integrated sensing devices, such as a 3D fingertip-shaped sensor BaroDome we validated in our manuscript.*”

That said, we would also like to highlight the continuity and evolution of our research. In our previous work [27], we developed a series of tactile sensors with increasing dimensional complexity: three 0D sensors (each incorporating one sensing modality—barometric unit, strain gauge, and accelerometer), a 1D sensor with six barometric units, and a 2D sensor with a 5×5 array of barometric units. In that study, we analyzed the taxel-value-isoline behaviors of different modalities and validated our theoretical model under pure normal force conditions. Specifically, our prior theory predicted a 187-fold super-resolution enhancement for the 1D sensor. A practical implementation machine learning (ML) model based on that theory achieved a 106-fold improvement for the 1D sensor and a 1260-fold improvement for the 2D sensor. In the current work, we extend this framework to 3D sensing and achieve a 1209-fold improvement with the 3D BaroDome sensor under normal forces. The main contribution of this manuscript is the modeling and understanding of the inherent systematic loss introduced by shear forces. Our theory predicts an average resolution loss of 0.33 mm due to shear effects, which provides a theoretical upper bound. The measured performance of BaroDome under shear shows an actual loss of 0.5 mm, consistent with our theoretical expectations. The sensor iterations are shown in the following figure.

To account for these effects, we extended our super-resolution model by incorporating shear force components. Through analysis of taxel isoline deformation, we observed that shear forces cause a systematic degradation in sensing performance, beyond what is seen with pure normal forces. This enhanced model enables the prediction of performance loss due to shear forces before sensor fabrication. To validate this prediction, we built the 3D BaroDome sensor and evaluated its performance under both pure normal and combined normal–shear force conditions. Experimental results confirmed our theoretical predictions: the sensing performance degrades systematically under shear, consistent with the model.

We believe this represents a closed-loop validation of our theoretical framework. Moreover, the model is generalizable to other sensor geometries, provided the sensor follows a grid-based configuration.

Additionally, in the Theory-Informed Tactile Sensor Design section, the statement “These design choices were informed by the theoretical analysis mentioned above” would be clearer if it explicitly specified the design choices and theoretically optimal performance indicated by the model for guiding the design of the current Barodome. The extent to which the performance and accuracy of Barodome align with the model’s predictions should be thoroughly discussed, as this is crucial for assessing the validity of the model in 3D-shaped sensor design.

This is a valid suggestion. We have incorporated clarifying text to enhance the readability of the procedure.

“More explicitly, we follow a systematic procedure to design and build the sensor prototype, comprising the following steps:

Step 1: Sensing Unit Selection We begin by selecting and analyzing a sensing unit embedded in a soft elastomer. Options include barometric sensors (for pressure), strain gauges (for elongation), and accelerometers (for inclination), as studied in our prior work [27]. We choose a barometer for its monotonic response and clear force–displacement mapping described by TVIs.

Step 2: Structural Design We examine the impact of elastomer thickness and the sensing unit’s position. Based on performance trade-offs (shown in Fig. 2), we select a 7.2 mm-thick elastomer with the sensor placed centrally.

Step 3: Material Selection We assess how material properties (Young’s modulus and Poisson’s ratio) influence TVIs. Following the analysis in [27], we use Smooth-On EcoFlex 00-30 (Young’s modulus: 0.07 MPa; Poisson’s ratio: 0.49999).

Step 4: Indenter Geometry Evaluation We evaluate how different indenter radii affect TVI shape, providing insights into expected performance across interaction scenarios. These indenters represent a wide range of objects that form single-point contact with the sensor. As discussed in [27], superresolution theory enables inference of the resultant force and contact center for single-point contacts. In the case of multiple contacts, they must be separated by at least one taxel spacing to be distinguishable. Continuous (contour) contact scenarios are beyond the current scope of our superresolution model.

Step 5: Shear Force Effects We extend our previous analysis to include shear forces, which are shown to systematically degrade accuracy (Fig. 3). This step reveals the shear-induced shift in TVI shape, essential

for predicting real-world performance loss.

Step 6: Noise Characterization We measure the sensor's intrinsic noise to determine the theoretical accuracy limit achievable via signal processing (Fig. 4, Fig. 5(d), and Fig. 6b).

Steps 1—5 define the TVI shape and its spatial characteristics—perception field, sensitivity, and attenuation profile—all of which govern the achievable resolution and accuracy. These parameters along with the sensor noise level measured from Step 6 allow us to theoretically estimate performance bounds, as shown in Fig. 2 of [27] and Fig. 6.

In prior work [27], we demonstrated: (1) a 187-fold theoretical resolution improvement for a 1D sensor, (2) a 106-fold improvement via machine learning, and (3) a 1260-fold improvement for a 2D sensor. In this work, we extend the analysis to a 3D fingertip-shaped sensor (Barodome), focusing on accuracy degradation evaluation due to shear forces. Our model predicts an inherent error of 0.33 mm under shear, and experimental results show a 0.5 mm loss—closely matching the theoretical prediction and validating the extended model.”.

Finally, it would be great to include a video demonstration showcasing Barodome's real-time operation.

We fully agree with the reviewer's suggestion that including a video demonstration could enhance the visibility and impact of our manuscript. We attempted to organize such an experiment at the Max Planck Institute for Intelligent Systems in Germany, where the study was primarily conducted. However, all authors have since relocated to new institutions for academic appointments.

Due to international travel and visa issues, as well as limited availability during the academic semester, coordinating this effort has become non-trivial. Several key authors have recently started tenure-track positions at different institutions, including one now based in China, further complicating logistics and visas.

We have had multiple discussions within the team to explore feasible solutions and to evaluate the cost-benefit of producing the video. At this stage, we believe the current materials presented in the manuscript are sufficient to support our primary claims.

We would greatly appreciate any further suggestions you may have regarding this point.

Answers to Reviewer 2

We sincerely appreciate your constructive feedback and rigorous check about our response and revision, which have significantly contributed to improving the structure and clarity of our manuscript.

I am reviewing this paper for the second time. In the first round, I was reviewer 2. The authors have thoroughly addressed many of the comments from all 3 reviewers, including adding a table comparing the sensor and superresolution results to other works, elaborating and better explaining many sections, adding raw signal plots, adding additional plots to figure 2, and adding a number of additional relevant references. I feel these changes have adequately addressed the comments.

Thank you very much for your careful and thorough review. We are truly grateful for the opportunity to benefit from such a high-quality reviewing experience.

My only remaining comment is that in the added plots in figure 2d, the rebuttal document says that these show "surface pressure versus lateral positions", however the actual figure 2d is labeled as force vs position. That said, if these are force vs position plots, then some of the curves do not match up with fig. 2c in the way that I expected. For example, the dark blue (10kPa, D=20) curve in 2d does not visually match the dark blue (D6, 10kPa) curve in 2c. Likewise, the dark blue (0.5kPa, D=20) curve in 2d does not visually match the red (D6, 0.5kPa) curve in 2c. So, either the label or the data seems to be wrong here.

We carefully reviewed your comments alongside our code and implementation, and we identified an error in the code where some curves were not correctly updated. We have corrected the issue and updated Fig.2d accordingly. The revised code is now available at code.

Incorrect Plot

Corrected Plot

Answers to Reviewer 3

Although current superresolution-based sensors (Barodome) are limited to resultant force and struggle to analyze local force distribution information on surfaces of complex-shaped objects, it is still necessary to experimentally validate their perception boundaries for such objects. It is recommended to supplement the "Results" section with qualitative test results for these objects (e.g., cubes or prisms), such as inferring contact regions through superposition of multi-indenter response models. Even if such experiments are preliminary explorations, they could demonstrate the authors' systematic delineation of the sensor's applicability boundaries and provide theoretical guidance for future improvements (e.g., integrating dense taxel layouts).

Thank you for your thoughtful comment. To address your concern, we have added the following text to the revised manuscript: *"Additionally, we would like to emphasize that the current theory is specifically developed for single-point contact inference with superresolution capabilities. Scenarios involving continuous or distributed contact patterns, such as those encountered with complex surface contours, fall outside the assumptions of our current model. Addressing such cases would require fundamentally different approaches, such as vision-based tactile sensors [20] or dense array-based tactile systems [35]."*

Scope of the current study

We appreciate the suggestion to perform qualitative tests with complex shapes (e.g., cubes or prisms) aimed at **contact regions inference**. However, those experiments are beyond this paper's scope. Our objective is to establish a **theoretical framework for tactile superresolution**, not to solve object-classification or full contact region inference. Inferring contact regions on complex geometries remains an open challenge because reliable ground-truth measurements are lacking, and any auxiliary sensor placed on Barodome would disturb its native behavior. Moreover, identifying an object after a single tap generally requires continuous contact data guided by a touching policy, which is not our current focus.

Key contributions and validation

Our theory unifies the variables that govern superresolution, including transduction methods, structural and material properties, object curvature, force amplitude/direction, and contact distance, via the Taxel Value Isoline (TVI). Using the TVI, we combine triangulation with an SNR-based information-theoretic metric to estimate the contact center and resultant force vector (Fig. 4). To validate the theory, we varied indenter radius over a wide (1 mm, 2 mm, 4 mm, 6 mm, 8 mm, 10 mm, and a flat surface), physically meaningful range (**Object Size and Evaluation sections; Fig. 2c, Fig. 6d**). Note that our Barodome has a radius of 10.75 mm. Radius provides a clean, interpretable proxy for curvature, allowing us to probe sharp-to-broad contacts without the ambiguity introduced by arbitrary shapes.

Applicability boundaries

As shown in [27], our model resolves single-point contacts; multiple contacts must be separated by at least one taxel spacing to be distinguished. Continuous or distributed contact patterns therefore demand a fundamentally different framework, pointing to vision-based or dense-array tactile sensors.

We hope this clarifies our rationale for the manuscript's focus and experimental design. We believe the presented results rigorously validate the proposed theory within its intended boundaries.